# Temporal Representations for Exploration: Learning Complex Exploratory Behavior without Extrinsic Rewards

**Faisal Mohamed**[1,2]     **Catherine Ji**[3]     **Benjamin Eysenbach**[3,*]     **Glen Berseth**[1,2,*]

[1]Mila-Quebec AI Institute, [2]Université de Montréal     [3]Princeton University

`faisal.mohamed@mila.quebec`, `cj7280@princeton.edu`

## Abstract

Effective exploration in reinforcement learning requires not only tracking where an agent has been, but also understanding how the agent perceives and represents the world. To learn powerful representations, an agent should actively explore states that contribute to its knowledge of the environment. Temporal representations can capture the information necessary to solve a wide range of potential tasks while avoiding the computational cost associated with full state reconstruction. In this paper, we propose an exploration method that leverages temporal contrastive representations to guide exploration, prioritizing states with unpredictable future outcomes. We demonstrate that such representations can enable the learning of complex exploratory x in locomotion, manipulation, and embodied-AI tasks, revealing capabilities and behaviors that traditionally require extrinsic rewards. Unlike approaches that rely on explicit distance learning or episodic memory mechanisms (e.g., quasimetric-based methods), our method builds directly on temporal similarities, yielding a simpler yet effective strategy for exploration.[1]

## 1 Introduction

Exploration remains a key challenge in reinforcement learning (RL), especially in tasks that demand reasoning over increasingly long horizons (Thrun, 1992) or with high-dimensional observations (Stadie et al., 2015; Burda et al., 2019b; Pathak et al., 2017).

Effective exploration in high-dimensional settings requires that agents (futilely) do not attempt to visit every last state, but only visit those states where they have something to learn. But how can an RL agent recognize such states? One direction is to leverage representation learning to compress the observations into a meaningful space where the agent can measure some sense of "usefulness," to drive and guide exploration. This raises the question *Which representations should be used to drive exploration?*

We start by observing that the RL problem is fundamentally about time, so representations that reflect temporal structure should be more useful than those that additionally include all bits required to reconstruct the input. We therefore adopt representations acquired by temporal contrastive learning. Theoretically, such representations are appealing because they are sufficient statistics for any Q function (Mazoure et al., 2023) (they are effectively a kernelized successor representation (Dayan, 1993; Barreto et al., 2017)). Computationally, these representations avoid the computational costs associated with world models and reconstruction (Achiam & Sastry, 2017; Stadie et al., 2015; Sekar et al., 2020; Bai et al., 2020). Indeed, prior work has shown that such representations are useful for learning policies (Myers et al., 2025) and value functions (Laskin et al., 2020). In our method, we use these representations to reward the agent for visiting states with unpredictable futures.

Our work is closely related to Jiang et al. (2025), which uses contrastive learning to estimate a similarity metric for exploration via quasimetric learning and constructs a reward signal using an episodic memory. Our method differs by (1) avoiding quasimetric learning and (2) avoiding episodic

---

[*]Equal advising.

[1]Project website: https://temp-contrastive-explr.github.io/

Figure 1: **Curiosity-Driven Exploration via Temporal Contrastive Learning.** We learn temporal representations so that the representation of $(s_0, a_0)$ is more similar to $(s_{2,3,4,...})$. We reward the agent for visiting future states that seem unpredictable. For example, from state $s_0$, state $s_1$ should confer lower reward than the state $s_4$.

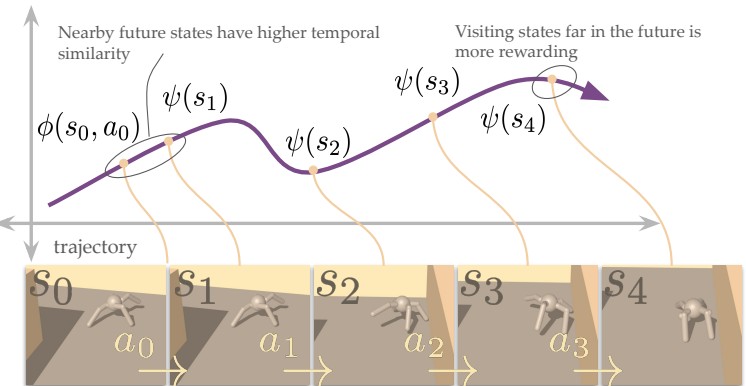

memory, which makes our method more amenable to off-policy RL algorithms. A graphical summary of our method is shown in Fig. 1.

The main contribution of this work is a new objective for exploration based on the prediction error of temporal representations. We demonstrate our approach by maximizing these intrinsic rewards with PPO (Schulman et al., 2017) and SAC (Haarnoja et al., 2018c). Our approach achieves state-of-the-art state coverage across navigation (`Ant` and `Humanoid` mazes), manipulation, and open-world environments (`Craftax-Classic`).

## 2 RELATED WORK

**Exploration.** Prior work on exploration (Schmidhuber, 2010; sch, 1991; Sorg et al., 2012; Brafman & Tennenholtz, 2002; Kearns & Singh, 2002) in reinforcement learning has proposed a variety of task-agnostic methods for encouraging agents to acquire diverse behaviors without relying on external rewards(also konwn as unsupervised RL) (Laskin et al., 2021). A central line of research focuses on intrinsic motivation, where agents seek novelty by maximizing state coverage or surprise. In low-dimensional and/or discrete environments, count-based exploration methods (Gardeux et al., 2016; Bellemare et al., 2016; Tang et al., 2017; Ostrovski et al., 2017; Martin et al., 2017; Xu et al., 2017; Machado et al., 2020) have demonstrated effective performance, particularly in Atari games. However, these methods often struggle in high-dimensional or continuous state spaces. In such settings, prediction-error-based approaches (Pathak et al., 2017; Burda et al., 2019a;b; Lee et al., 2019; Guo et al., 2022) have been more successful, showing effectiveness both in video game environments and continuous control tasks. Another direction leverages representation learning: compact features are extracted from raw inputs, and entropy estimators are applied to these representations to quantify novelty (Liu & Abbeel, 2021; Laskin et al., 2022).

Beyond novelty-driven exploration, another class of methods emphasizes the agent's ability to influence or regulate its environment. Empowerment-based approaches maximize mutual information between states and actions, encouraging agents to discover actions that yield significant control over future states (Klyubin et al., 2005b;a; Biehl et al., 2015; Zhao et al., 2021; Mohamed & Jimenez Rezende, 2015; Karl et al., 2019; Hayashi & Takahashi, 2025; Levy et al., 2024; Jung et al., 2011; Du et al., 2020; Myers et al., 2024). While conceptually appealing, solving the full empowerment objective remains intractable. A complementary perspective is surprise minimization, where agents reduce prediction uncertainty to maintain stability or create structured niches in the environment (Friston, 2010; Berseth et al., 2021; Rhinehart et al., 2021; Hugessen et al., 2024). These approaches demonstrate how regulating predictability can give rise to complex behaviors in both fully and partially observed domains.

**Representation learning for RL.** Prior work on representation learning for RL focuses on self-supervised methods to improve the data efficiency of RL agents. A notable approach in this category involves the use of unsupervised auxiliary tasks, where a pseudo-reward is added to the task reward to shape the learned representations and provide an additional training signal. Examples of this approach include (Jaderberg et al., 2017; Farebrother et al., 2023; Oord et al., 2018; Laskin et al., 2020; Schwarzer et al., 2021). Another line of work focuses on forward-backward representations (Touati &

Ollivier, 2021; Touati et al., 2023), which aim to capture the dynamics under all optimal policies and have been shown to exhibit zero-shot generalization capabilities. Moreover, contrastive learning has been applied in various exploration settings, including goal-conditioned learning (Eysenbach et al., 2022; Liu et al., 2025), skill discovery (Laskin et al., 2022; Yang et al., 2023; Zheng et al., 2025), and state coverage or curiosity (Liu & Abbeel, 2021; Du et al., 2021; Yarats et al., 2021). In the context of curiosity-driven exploration, (Du et al., 2021; Yarats et al., 2021) employ contrastive learning to learn visual representations in image-based environments, where the RL agent is trained to maximize the error of the representation learner (similar in spirit to prediction-error approaches), Our work is similar to these method as it also uses contrastive learning, but for learning representations that capture the temporal structure of the policy and environment dynamics, without explicit world-modeling, or skill learning. Jiang et al. (2025) uses a special parametrization of contrastive learning to learn temporal distances via quasimetric learning. It then constructs an aggregated intrinsic reward to maximize the minimum temporal distance between the state at the current time step and the states from previous time steps, which are stored in an episodic memory. Our work is closely related to Jiang et al. (2025), which likewise uses contrastive learning to estimate a similarity metric for exploration. Our method differs by (1) avoiding the quasimetric parametrization and (2) avoiding episodic memory, which makes our method more amenable to the off-policy setting. We compare with ETD (Jiang et al., 2025) in the experiments.

## 3 BACKGROUND

We consider a controlled Markov process , defined by time-indexed states $s_t$ and actions $a_t$. The initial state is sampled from $p_0(s_0)$, and subsequent states are sampled from the Markovian dynamics $p(s_{t+1} \mid s_t, a_t)$. Actions are selected by a stochastic, parameterized policy $\pi(a_t \mid s_t)$. Without loss of generality, we assume that episodes have an infinite horizon; the finite-horizon problem can be incorporated by augmenting the dynamics with an absorbing state. The key to C-TeC is to use a self-supervised, or intrinsic reward, built on temporal contrastive representations. We detail the necessary preliminaries below.

**Discounted state occupancy measure.** Formally, we define the $\gamma$-discounted state occupancy measure of policy $\pi$ conditioned on a state and an action (Ho & Ermon, 2016; Eysenbach et al., 2021; 2022) as

$$p_\pi(s_f \mid s_t, a_t) \triangleq (1-\gamma) \sum_{\Delta=0}^{\infty} \gamma^\Delta p_\pi(s_{t+\Delta} = s_f \mid s_t, a_t), \tag{1}$$

where $p_\pi(s_f \mid s, a)$ is the probability of being at future state $s_f$ conditioned on $s_t, a_t$ and following policy $\pi$. In continuous settings, the future state distribution $p_\pi(s_f \mid s, a)$ is a probability *density*.

Traditionally, the discounted state occupancy measure is defined with respect to a policy as $p_\pi(s_f \mid s_t, a_t)$. However, in this work, the intrinsic reward $r_{\text{intr}}$ is defined using a discounted state occupancy measure over the trajectory buffer $\mathcal{T}$, which contains trajectories collected from a history of policies:

$$p_\mathcal{T}(s_f \mid s_t, a_t) \triangleq (1-\gamma) \sum_{\Delta=0}^{\infty} \gamma^\Delta p_\mathcal{T}(s_{t+\Delta} = s_f \mid s_t, a_t).$$

To sample from the *trajectory buffer* distribution $p_\mathcal{T}(s_f \mid s_t, a_t))$, we first sample an offset $\Delta \sim \text{GEOM}(1-\gamma)$, then set the future state $s_f = s_{t+\Delta}$. Here, future state $s_f = s_{t+\Delta}$ is the state reached from $(s_t, a_t)$ after executing $\Delta$-number of actions within a sampled stored trajectory.

**Contrastive learning.** Contrastive representation learning methods (Chopra et al., 2005; Oord et al., 2018; Chen et al., 2020) train a critic function $C_\theta$ that takes as input pairs of positive and negative examples, and learn representations so that positive pairs have similar representations and negative pairs have dissimilar representations. To estimate the discounted state occupancy, positive examples are sampled from a joint distribution $p_\mathcal{T}((s_t, a_t), s_f) = p_\mathcal{T}(s_t, a_t)p_\mathcal{T}(s_f \mid s_t, a_t)$, while the negative examples are sampled from the product of marginal distributions $p_\mathcal{T}(s_t, a_t)p_\mathcal{T}(s_f)$. Here, $p_\mathcal{T}(s_f)$ is the marginal discounted state occupancy $p_\tau(s_f) = \iint p_\tau(s_f \mid s_t, a_t)p_\mathcal{T}(s_t, a_t) \; ds_t \, da_t$. We use the InfoNCE loss to train the contrastive learning model (Oord et al., 2018). Let $\mathcal{B} = \{(s_t^{(i)}, a_t^{(i)}, s_f^{(i)})\}_{i=1}^K$ be the sampled batch, where $s_f^{(1)}$ is the positive example and $\{s_f^{(2:K)}\}$ are the $K-1$ negatives sampled independently from $(s_t^{(i)}, a_t^{(i)})$. In addition to the standard InfoNCE

objective, prior work has shown that a LogSumExp regularizer is necessary for control (Eysenbach et al., 2021). The full contrastive reinforcement learning (CRL) loss is as follows:

$$\mathcal{L}_{\text{CL}}(\theta) = -\mathbb{E}_{\substack{(s_t,a_t)\sim p_{\mathcal{T}}(s_t,a_t) \\ s_f^{(1)}\sim p_{\mathcal{T}}(s_f|s_t,a_t) \\ s_f^{(2:K)}\sim p_{\mathcal{T}}(s_f)}} \left[ \log\left( \frac{e^{C_\theta((s_t,a_t),s_f^{(1)})/\tau}}{\sum_{j=1}^{K} e^{C_\theta((s_t,a_t),s_f^{(j)})/\tau}} \right) \right]. \quad (2)$$

where $\tau$ is a temperature parameter. The optimal critic $C^*((s_t,a_t),s_f)$ corresponds to a log probability ratio (Ma & Collins, 2018), $C^*((s_t,a_t),s_f) \approx \log p_{\mathcal{T}}(s_f \mid s_t,a_t) - \log p_{\mathcal{T}}(s_f)$, where we use the negative $\ell^1$ and $\ell^2$ distances as the critic function (see Appendix G.3) Conceptually, the critic $C_\theta$ gives a temporal similarity score between state-action pairs $(s_t,a_t)$ and future states $s_f$ via learned representation $\phi_\theta$ and $\psi_\theta$.

# 4 EXPLORATION VIA TEMPORAL CONTRASTIVE LEARNING

To improve exploration, we learn representations that encode the agent's future state occupancy using temporal contrastive learning. We begin by describing how contrastive representation learning can be used to estimate state occupancy by learning a similarity function that assigns high scores to frequently visited future states and low scores to rarely visited ones (Eysenbach et al., 2022; Oord et al., 2018). We then explain how this similarity score can be leveraged to derive an intrinsic reward signal for exploration.

## 4.1 TRAINING THE CONTRASTIVE MODEL

The contrastive model $C_\theta(s_t,a_t,s_f)$ is trained on batches $\mathcal{B}$ of $(s_t,a_t,s_f)$ tuples, where each $s_f$ is sampled from the discounted future state distribution (Section 3). We use two parameterized encoders to define the contrastive model: $\phi_\theta(s_t,a_t)$ for state-action pairs and $\psi_\theta(s_f)$ for future states. A batch of state-action pairs $\{(s_t^{(i)},a_t^{(i)})\}_{i=1}^K$ is passed through $\phi_\theta$, while the corresponding batch of future states $\{s_f^{(i)}\}_{i=1}^K$ is passed through $\psi_\theta$. The resulting representations are then normalized to have unit norm. To compute the similarity between representations in practice, we found that using either the negative $\ell^1$ or $\ell^2$ norm was effective, depending on the environment. The contrastive encoder is trained to minimize the InfoNCE loss (Equation (2)) (Oord et al., 2018). For each batch sample, the positive examples of other samples are treated as negatives, following common practice (Chen et al., 2020). The temperature parameter $\tau$ is learned during training. The details of the implementation are provided in Appendix E.

## 4.2 EXTRACTING AN EXPLORATION SIGNAL FROM THE CONTRASTIVE MODEL

Given the contrastive model, a useful intrinsic reward can be constructed. The aim is to reach unexpected but *meaningful* states. This is in contrast to surprise maximization or similar objectives which may prioritize unexpected but meaningless (i.e. random) states as those observed in the Noisy TV problem (see Figure 21) (Gruaz et al., 2024).

The contrastive model produces a similarity score between state-action pairs $(s_t,a_t)$ and future states $s_f$. *Negating* this similarity score results in our exploration signal $r_{\text{intr}}$, encouraging the agent to visit states that are not predictive of future states in the same trajectory *in the eyes of the representations*. The expression for the expectation of $r_{\text{intr}}$ is as follows:

$$\mathbb{E}[r_{\text{intr}}(s_t,a_t)] = \mathbb{E}_{p_{\mathcal{T}}(s_f|s_t,a_t)}[-C_\theta((s_t,a_t),s_f)] = \mathbb{E}_{p_{\mathcal{T}}(s_f|s_t,a_t)}[||\phi_\theta(s_t,a_t) - \psi_\theta(s_f)||] \quad (3)$$

where the norm can be taken to be $\ell^1$ or $\ell^2$ (See Section 6). Here, $r_{\text{intr}}$ rewards the agent for exploring states that provide the least amount of information about future states. The reward captures both temporal distance and possible inconsistencies in the model, where the representations assign erroneously low relative likelihoods to future states (see Section 5.2 for analysis).

We use PPO (Schulman et al., 2017) and SAC (Haarnoja et al., 2018b;a;c) for policy training (pseudocode in Algorithm 1). In practice, we found that using a single sample future state to approximate the expectation in Equation (3) works well, except in Craftax-Classic, where we used a

---

**Algorithm 1** Curiosity-Driven Exploration via Temporal Contrastive Learning

---

1: Initialize: $\pi, \phi_\theta, \psi_\theta$, trajectory buffer $\mathcal{T}$
2: **for** each iteration **do**
3:     **for** each environment step $1 \leq t \leq T$ **do**
4:         $a_t \sim \pi(a_t \mid s_t), s_{t+1} \sim p(s_{t+1} \mid s_t, a_t),$
5:         $\tau_j \leftarrow \tau_j \cup \{s_t, a_t, s_{t+1}\},$
6:     $\mathcal{T} \leftarrow \mathcal{T} \bigcup \tau_j, \tau_j \leftarrow \{\}$
7:     Sample $\{(s_t^i, a_t^i)\}_{i=1}^{|\mathcal{B}|} \sim \mathcal{T}$               ▷ Sample a batch of state,action pairs
8:     Sample $\Delta_i \sim \text{GEOM}(1 - \gamma) \, \forall i \in \{1, 2, \ldots, |\mathcal{B}|\}$      ▷ Sample a geometric offsets
9:     Set $s_f^i = s_{t+\Delta_i}^i \, \forall i \in \{1, 2, \ldots, |\mathcal{B}|\}$     ▷ Set the future state $s_{f_i}$ according to $\Delta_i$
10:    Compute intrinsic rewards: $\mathbf{r}_i = -C_\theta((s_t^i, a_t^i), s_f^i)$              ▷ Equation (3)
11:    Update representations: $\theta \leftarrow \theta - \eta \nabla_\theta \mathcal{L}_{\text{InfoNCE}}(\mathcal{B} = \{(s_t^i, a_t^i, s_f^i)\}_{i=1}^{|\mathcal{B}|}; \theta)$     ▷ Equation (2)
12:    RL update using $\{(s_t^i, a_t^i, \mathbf{r}_t^i)\}_{i=1}^{|\mathcal{B}|}$         ▷ Update the policy using PPO/SAC

---

Monte Carlo estimate. Additional details are provided in Appendix E.4.1, furthermore, we ablate the future state sampling strategy. We observe that sampling from the discounted occupancy measure yields good performance across environments and we stick to this strategy in our experiments. We also show the performance differences between these sampling strategies in Appendix G.6.

## 5 INTERPRETATION OF C-TEC

In the below sections, we provide intuition for how the representation-parameterized intrinsic reward may drive effective exploration behavior. Sec. 5.1 details an info-theoretic interpretation of C-TeC, ignoring learned representations to help build intuition and compare with other common objectives (see Appendix I). In Section 5.2, we discuss the importance of the learned contrastive representations to C-TeC performance. For example, contrastive representations enable C-TeC's performance to remain the same with or without noise in the Noisy TV environment (Figure 21).

### 5.1 INFORMATION-THEORETIC EXPRESSION OF C-TEC

The intrinsic reward has an information-theoretic interpretation. We consider the limit where representations perfectly capture the underlying point-wise mutual information (MI). In this regime, the intrinsic reward evaluates to the negative of the KL-divergence between the conditional future-state distribution $p_\mathcal{T}(s_f \mid s_t, a_t)$ and the marginal future-state distribution $p_\mathcal{T}(s_f)$:

$$\mathbb{E}[r_{\text{intr}}(s_t, a_t)] = -\mathbb{E}_{p_\mathcal{T}(s_f|s_t,a_t)}\left[\log \frac{p_\mathcal{T}(s_f \mid s_t, a_t)}{p_\mathcal{T}(s_f)}\right] \tag{4}$$

$$= -D_{\text{KL}}[p_\mathcal{T}(s_f \mid s_t, a_t) \,\|\, p_\mathcal{T}(s_f)]. \tag{5}$$

This intrinsic reward describes mode-seeking behavior (Murphy, 2022, Section 6.2.6): the conditional should only have support where the marginal $p_\mathcal{T}(s_f)$ has support. This optimization is distinct from minimizing the forward KL-divergence $D_{\text{KL}}[p_\mathcal{T}(s_f) \,\|\, p_\mathcal{T}(s_f \mid s_t, a_t)]$, which instead prioritizes mean-seeking behavior over regions of the state space where the marginal may *not* have support.

This mode-seeking reward can be interpreted as prioritizing $(s, a)$ that are minimally informative about reached future states:

$$\mathbb{E}[r_{\text{intr}}(s_t, a_t)] = -D_{\text{KL}}[p_\mathcal{T}(s_f \mid s_t, a_t) \,\|\, p_\mathcal{T}(s_f)]$$
$$= \underbrace{H[S_f \mid s_t, a_t]}_{\text{surprise}} + \underbrace{\mathbb{E}_{p_\mathcal{T}(s_f|s_t,a_t)}[\log p_\mathcal{T}(s_f)]}_{\text{``familiarity'' term}},$$

where $S_f$ denotes the future state *random variable* and $s_f \sim p_\mathcal{T}(s_f \mid s_t, a_t)$. State-action pairs with spread-out trajectories ("surprise") over states that *have actually been seen* ("familiarity") have higher reward i.e., a high reward should be given to states that have been visited but are temporally distant, rather than giving a high reward to unvisited states.. States encountered during roll-out are then added to the marginal, and the process repeats.

The success of the intrinsic reward cannot solely be attributed to "fitting" $p_{\mathcal{T}}(s_f \mid s, a)$ to $p_{\mathcal{T}}(s_f)$ via the policy rollout distribution. To test the hypothesis that the mode-seeking behavior is important, we ran experiments where the intrinsic reward is the forward mean-seeking KL (Equation (20)). Appendix G.5 shows that the objective succeeds because it is minimizing this mode-seeking formulation of the KL rather than fitting the conditional future states to a broad marginal. A linear stability analysis on the fixed points of C-TeC is in Appendix J.1; we simplify the problem setting for analysis and find that there are no easily-achievable stable fixed points for general nontrivial MDPs, meaning that the distribution over reached states continually evolves with iteration time.

## 5.2 REPRESENTATIONS ARE NECESSARY FOR C-TEC TO SUCCEED

The representations not only capture a raw info-theoretic exploration signal but also a form of representation prediction error. All of the analysis in Section 5.1 assumes a fully-expressive critic that perfectly captures the point-wise MI. However, the true learned representations only approximate the point-wise MI. The full expected intrinsic reward is the following

$$\mathbb{E}[r_{\text{intr},\phi,\psi}(s_t, a_t)] = -\mathbb{E}_{p_{\mathcal{T}}(s_f|s_t,a_t)}\left[\log \frac{p_{\phi,\psi}(s_f \mid s_t, a_t)}{p_{\phi,\psi}(s_f)}\right]$$

where $p_{\phi,\psi}$ describe probability ratios under the learned contrastive representations $\phi$ and $\psi$. Thus, the reward prioritizes exploration in areas where state-actions are not informative of future states *according to a contrastive model*. In other words, the method rewards state-actions with low predictability of future states. Learned representations that fail to capture features necessary for the classification task lead to higher intrinsic reward.

When the representations *do* capture features important for temporal classification, the resulting reward is invariant to classification-irrelevant perturbations such as spurious noise. This is a highly useful property: in the Noisy TV environment, C-TeC performance is strong (see Fig. 21 results) and unaffected by noise. Noise randomly sampled from the same distribution every timestep does not lead to stronger classifier performance, and, thus, the intrinsic reward is invariant to these distractors.

Finally, the contrastive representations are crucial C-TeC's performance. Experimental results in Appendix G.4 show that the method is *not* robust to the usage of a monolithic critic $f(s, a, g)$, indicating that the representation parameterization of the critic is key.

## 6 EXPERIMENTS

Our experiments show that contrastive representations can be used to reward the agent for visiting less-occupied or distant future states. We then use the C-TeC reward function for exploration in robotic environments and Craftax-Classic. We mainly study the following questions: (**Q3**) How well does C-TeC compare to ETD Jiang et al. (2025)? (**Q2**) How well does C-TeC reward capture the agent's future state distribution? (**Q3**) How effectively does C-TeC explore in locomotion, manipulation, and Craftax environments compared to prior work?

**Environments** We use environments from the JaxGCRL codebase (Bortkiewicz et al., 2025). Specifically, we evaluate C-TeC on the `ant_large_maze`, `humanoid_u_maze`, and `arm_binpick_hard` environments, which require solving long directed plans to reach goal states. In the maze-based environments, the agent's objective is to reach a designated goal specified at the start of each episode. Exploration in these settings corresponds to maze coverage: an agent

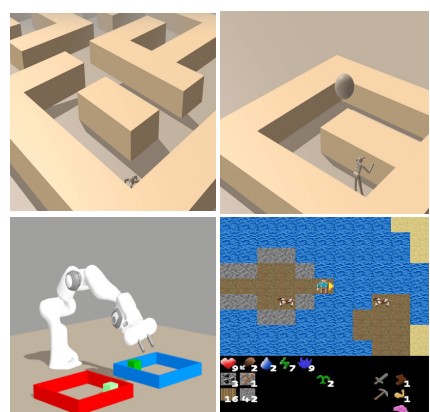

Figure 2: **Environments**. Maze coverage, robotic manipulation, and the survival game Craftax.

that visits more unique positions in the maze demonstrates better exploration capabilities. In the `arm_binpick_hard` environment, which differs from the more navigation-themed tasks used in prior work, the agent must pick up a cube from a blue bin and place it at a specified target location in

a red bin. This represents a challenging exploration task, as the agent must locate the cube, grasp it, and successfully place it at the correct target location.

Our experiments with the `ant` and `humanoid` agents assess the method's ability to achieve broad state coverage using two complex embodiments. Meanwhile, the `arm_binpick_hard` task evaluates the method's effectiveness at exploration in an object manipulation setting. We also run C-TeC on Craftax-Classic (Matthews et al., 2024), a challenging open-world survival game resembling a 2D Minecraft. The agent's goal is to survive by crafting tools, maintaining food and shelter, and defeating enemies

In the locomotion and manipulation environments, we compare C-TeC to common prior methods for exploration: **Random Network Distillation (RND)** (Burda et al., 2019b) and **Intrinsic Curiosity Module (ICM)** (Pathak et al., 2017), both of which are popular intrinsic motivation methods for exploration. **Active Pre-training (APT)** (Liu & Abbeel, 2021): APT learns observation representations using contrastive learning, where positives are augmentations of the same observation and negatives are different observations. It uses the KNN distance between state representations as an exploration signal, which correlates with state entropy. Unlike C-TeC, APT does not learn representations that are predictive of the future.

In Craftax, we compare against RND, ICM, and **Exploration via Elliptical Episodic Bonuses (E3B)** (Henaff et al., 2022), a count-based exploration method. We found that using the negative $L_1$ distance (Equation (16)) as the critic function works best in the robotics environments, while the negative $L_2$ distance (Equation (17)) performs best in Craftax. A comparison of different critic functions can be found in the appendix. We also compare our method to ETD (Jiang et al., 2025). While C-TeC is implemented in JAX, to ensure a fair comparison we re-implemented it on top of the ETD codebase and ran the experiments on the same robotic environments as well as on Crafter (Hafner, 2022). The comparison results are presented in Section 6.1.

## 6.1 COMPARISON TO ETD (Q1)

In this section, we compare C-TeC to ETD (Jiang et al., 2025), a recent method that uses contrastive learning to learn a quasimetric that encodes temporal distances between states via metric residual networks (MRN) (Liu et al., 2023; Myers et al., 2024). ETD extracts an exploration signal by measuring the minimum temporal distance between the current state and previous states stored in an episodic memory. By contrast, C-TeC is simpler: it does not require episodic memory or an explicit quasimetric, and it works in both on-policy and off-policy settings.

For a fair comparison, we implemented C-TeC on top of the ETD codebase[2]. We ran C-TeC across multiple hyperparameter configurations (primarily ablating different contrastive similarity functions) and selected the configuration with the best overall performance. We used the same contrastive encoder architecture as ETD, and for ETD we adopted the hyperparameters reported in Appendix E of Jiang et al. (2025). Our experiments use only intrinsic rewards, as our goal is to understand C-TeC's behavior in the absence of any task rewards.

Figure 3 shows the results on the robotic environments and Crafter. All experiments are conducted in the intrinsic exploration setting (without providing the task reward). Both methods perform similarly on `ant_hardest_maze`. C-TeC outperforms ETD in `humanoid_u_maze`, albeit with higher variance, while ETD performs better in `arm_binpick_hard`. In Crafter, however, C-TeC significantly outperforms ETD.

We speculate that this improvement stems from a difference in the exploration signals: ETD encourages novelty by maximizing the minimum temporal distance from past states (backward-looking), while C-TeC prioritizes states that can lead to a larger set of possible future states (forward-looking). This forward-looking perspective may better capture the long-term exploratory value of states, which could explain C-TeC 's stronger performance in Crafter(We refer the reader to Appendix K and J.2 for a didactic example that illustrate the differnce between C-TeC and ETD rewards). Moreover, as mentioned, ETD requires a more constrained architecture, specifically the MRN (Liu et al., 2023), to learn quasimetric representations that encode temporal distance. Our findings show that such architectural constraints are not necessary for effective exploration. Learning representations that

---

[2]https://github.com/Jackory/ETD/tree/main

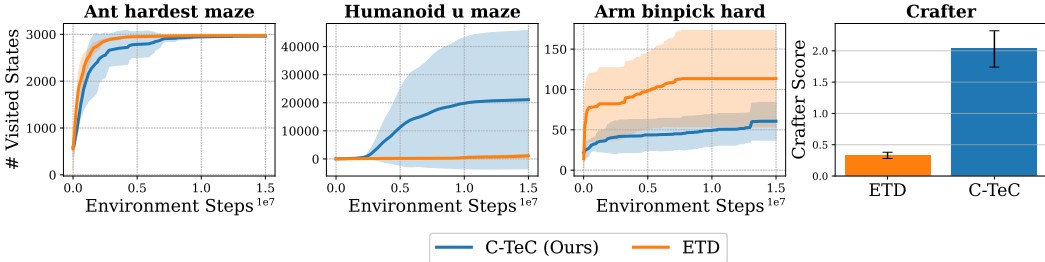

Figure 3: **C-TeC Performance compared to ETD** ([Jiang et al., 2025](#)) C-TeC is competitive to ETD in terms of state coverage in continuous control environments, and outperform ETD in Crafter.

capture the temporal structure of policy behavior and environment dynamics seems sufficient to achieve meaningful exploration without the MRN architecture or temporal distances.

The key takeaway is that, while C-TeC is conceptually similar to ETD, **C-TeC achieves comparable or stronger performance while also reducing algorithmic complexity**.

## 6.2 LEVERAGING THE FUTURE STATE DISTRIBUTION FOR EXPLORATION (Q2, Q3)

In this section we demonstrate that the C-TeC reward captures the future state distribution and it can be used to incentivize the agent to visit less-occupied and more distant future states. We visualize the C-TeC reward at different stages of training in the `ant_hardest_maze` environment. The contrastive critic is defined as the negative $L_1$ distance (Equation (16)), and the policy is trained to maximize the intrinsic reward defined in Equation (3). Figure 4 shows the reward values in a section of the maze. Following the qualitative results in Figure 4, we evaluate C-TeC in the `ant_large_maze`, `humanoid_u_maze`, and `arm_binpick_hard` environments. We run two variants of the experiment: (1) using the complete state vector as the future state, which is common in exploration tasks where the agent is encouraged to explore the entire state space; and (2) incorporating prior knowledge by narrowing the future state to specific components of the state vector. The latter allows us to assess whether C-TeC can flexibly explore subspaces of the state space, which is often useful in practice. In `ant_large_maze`, we define the future state as the future (x, y) position of the ant's torso. In `humanoid_u_maze`, we use the future (x, y, z) position of the humanoid's torso. Finally, in `arm_binpick_hard`, we define the future state as the future position of the cube.

As an evaluation metric, we count the number of unique discretized states covered by each agent. In `ant_large_maze`, we count the number of unique (x, y) positions in the maze visited by each agent. Similarly, in `humanoid_u_maze`, we count the number of visited (x, y, z) positions, and in `arm_binpick_hard`, we count the number of unique cube positions. We compare C-TeC to RND, ICM, APT, and a uniformly random policy. Figure 5 shows the learning curve when using the complete future state vector while Figure 10 shows the performance when we incorporate prior knowledge by restricting the future state to specific components of the state vector. . Each agent is run with 5 random seeds, and we plot the mean and standard deviation ([Patterson et al., 2024](#)).

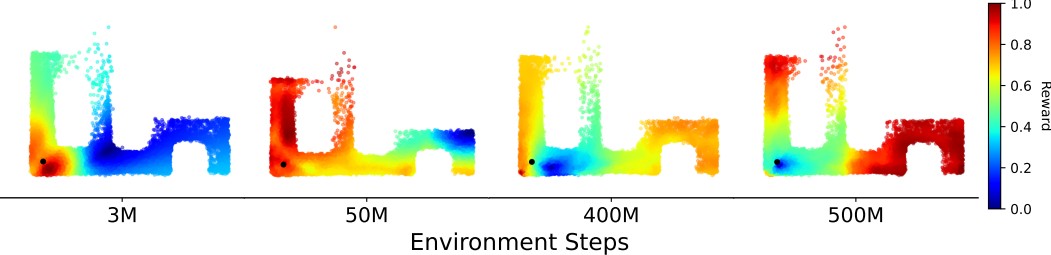

Figure 4: **Evolution of the C-TeC reward during training**. This figure shows how the intrinsic reward changes over the course of training based on future state visitation. The black circle in the lower-left corner represents the starting state. C-TeC reward captures the agent's future state density and rewards the agent for visiting states faraway in the future.

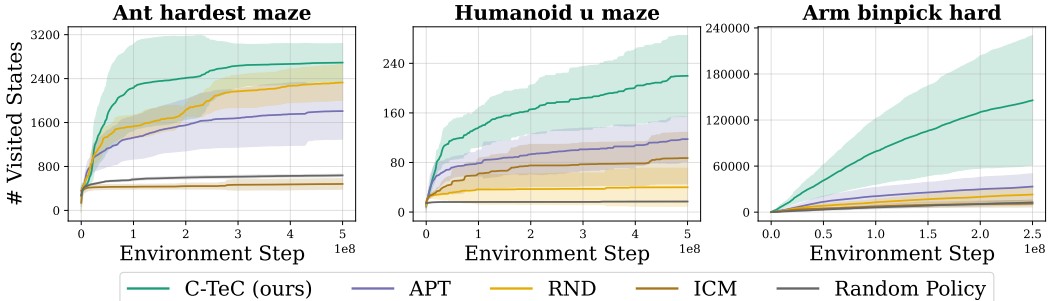

Figure 5: **C-TeC explores more states than prior methods.** We compare the state coverage of C-TeC to APT (Liu & Abbeel, 2021), RND (Burda et al., 2019b) and ICM (Pathak et al., 2017). We include a uniform random policy as well.

Our agent outperforms the baselines in both variants of the experiment and learns interesting behaviors in the challenging `humanoid_u_maze` environment. Figure 6 shows screenshots of C-TeC behavior. More visuals are provided in Appendix M. This improvement can be the result of C-TeC's consistent reward properties. Methods like RND and ICM will eventually tend to zero reward as the state distribution is covered. A nice property of C-TeC is that it does not have zero reward in the limit.

## 6.3 Learning complex behavior in Craftax-Classic

Can an RL policy learn complex behavior in Craftax-Classic without any task reward? To answer this question, we run C-TeC on Craftax-Classic (Matthews et al., 2024), a complex survival game where the agent's goal is to survive by crafting tools, maintaining food and shelter, and defeating enemies.

In this experiment, we use the same PPO implementation as used in the baselines in the Craftax paper (Matthews et al., 2024), adding the C-TeC reward on top of it.

We compare against RND, ICM, E3B, and a uniform random policy. We found that using PPO with memory (PPO-RNN) yields the best

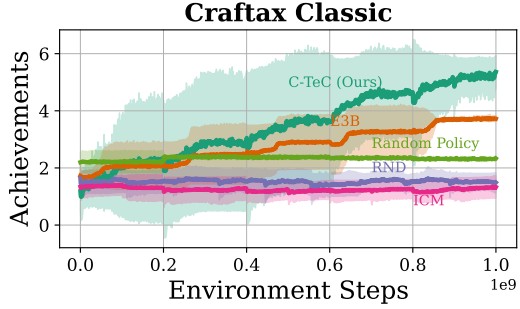

Figure 7: C-TeC achieves higher achievements in the Craftax survival game.

performance. The results are presented in Figure 7. The y-axis represents the sum of the achievements success rate, which measures how many capabilities and useful objects the agent has discovered. C-TeC outperforms the baselines and unlocks more achievements. Figure 29 visualizes some of the achievements of the C-TeC agent during an evaluation episode.

## 7 Conclusion

This work has shown how to learn and leverage temporal contrastive representations for intrinsic exploration. With these representations, we construct a reward function that seeks out states with unpredictable future outcomes. We find that C-TeC is a simple method that yields strong performance on state visitation metrics. These results hold across different RL algorithms and environments. Future work includes further investigating the role of temporal representations in effective exploration, combining the C-TeC reward with task rewards, and adapting the method to pixel-based and partially observed settings.

---

[2]Agent videos: https://temp-contrastive-explr.github.io/

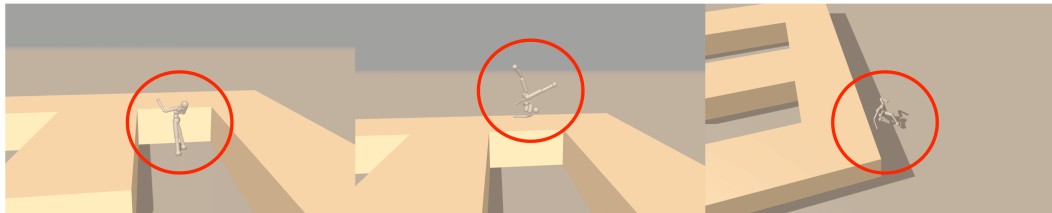

Figure 6: **C-TeC behavior in humanoid-u-maze**. C-TeC agent learns to escape the u-maze by jumping over the wall. None of the alternative exploration methods discovered this kind of unexpected behavior[2].

## REPRODUCIBILITY STATEMENT

For reproducing the paper's results, we provide the algorithm codebase in the supplementary material and in the GitHub link `https://github.com/FaisalAhmed0/c-tec.git` training details and hyperparameters are in Appendix E.

**Acknowledgments.** We thank Marco Jiralerspong and Daniel Lawson for feedback on the draft of the paper. We thank Daniel Lawson, Roger Creus Castanyer, Siddarth Venkatraman, Raj Ghugare, Mahsa Bastankhah, and Grace Liu on discussions throughout the project. We thank Liv d'Aliberti for their plotting code and format for Figure 28. We thank the anonymous reviewers for helpful comments and feedback that improved the paper. We want to acknowledge funding support from Natural Sciences and Engineering Research Council of Canada, Samsung AI Lab, Google Research, Fonds de recherche du Québec, The Canadian Institute for Advanced Research (CIFAR), and IVADO. We acknowledge compute support from Digital Research Alliance of Canada, Mila IDT, and NVIDIA.

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

## A  USAGE OF LARGE LANGUAGE MODELS (LLMS)

We used LLMs as a grammar and spelling correction tool. We provide each section in the prompt and ask the LLM to correct any obvious grammatical or spelling mistakes; however, we prevent the LLM from changing the style or introducing any new claims.

## B  SAMPLE EFFICIENCY OF C-TEC

In this section, we show the performance of C-TeC with different amounts of environment steps. In the main experiments, we used 500M environment steps. In Table 1, we present the results of running C-TeC with significantly fewer environment steps: 50M (10× less) and 30M (16× less) than the main experiments. Our results demonstrate that C-TeC can explore effectively with fewer environment interactions, and they also highlight C-TeC 's scalability with respect to the number of environment interactions, an important property for a pure exploration method. We also visualize the state coverage with the C-TeC reward heatmap in the ant-hardest-maze by plotting the x,y positions that the agent covered during training (Figure 8).

| Environment | 500M steps | 50M steps | 30M steps |
|---|---|---|---|
| Ant-hardest-maze | $2500 \pm 300$ | $1916 \pm 430$ | $1119 \pm 304$ |
| Humanoid-u-maze | $230 \pm 40$ | $143 \pm 34$ | $102 \pm 11$ |
| Arm-binpick-hard | $135000 \pm 10000$ | $40000 \pm 14000$ | $31150 \pm 3156$ |

Table 1: Sample Efficiency of C-TeC

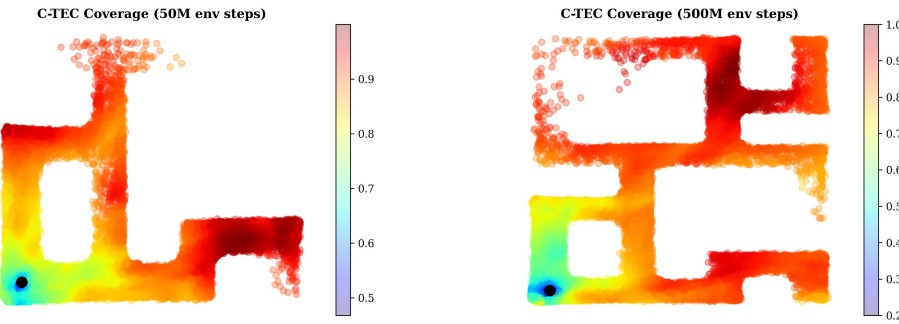

Figure 8: **C-TeC State Coverage** The Black circle in the lower left corner is the starting state. The figure on the left shows the state coverage when we run C-TeC with 50M environment steps in `ant-hardest-maze`. On the right, we show state coverage when we run C-TeC with 500M environment steps.

## C  DOES C-TEC'S CONTRASTIVE MODEL SUFFER FROM REPRESENTATION COLLAPSE?

In this section, we investigate the representations of C-TeC's contrastive model. Specifically, do the contrastive representations suffer from mode collapse? This is a common issue in contrastive learning, as one possible local optimum is to output a constant vector if the negative examples are not chosen carefully. As a measure for collapse, we plot the variance of the contrastive representations in the robotic environments during training in Figure 9. Our plot shows that C-TeC does not suffer from mode collapse, and the variance is steady during most of the training time.

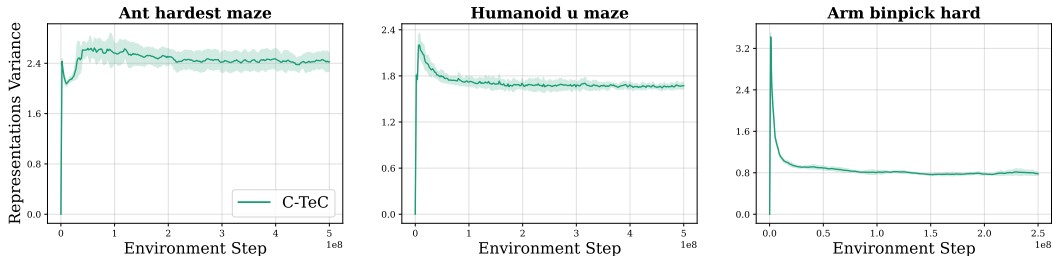

Figure 9: **Representations Variance** C-TeC's contrastive model does not suffer from mode collapse.

# D INCORPORATING PRIOR KNOWLEDGE

Figure 10 shows the performance when we incorporate prior knowledge on our method by restricting the future state to specific components of the state vector. . Each agent is run with 5 random seeds, and we plot the mean and standard deviation (Patterson et al., 2024).

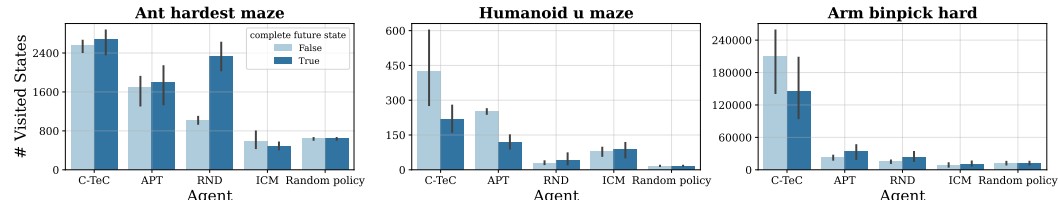

Figure 10: **State coverage when leveraging prior knowledge** C-TeC outperforms prior methods APT, RND, ICM, and can explore effectively when leveraging prior knowledge. This shows the improved flexibility of C-TeC in incorporating prior knowledge by narrowing the exploration space compared to prior work.

# E TRAINING DETAILS

We summarize the hyperparameters and model architectures for all experiments. In Appendix E.1, we provide the training details for the locomotion and manipulation experiments. In Appendix E.2, we provide the details of the Craftax experiments. In Appendix E.3, we provide the details of each environment.

Finally, in Appendix G, we include the ablation experiments.

## E.1 ROBOTICS ENVIRONMENTS

In the robotics environments, we used SAC as the RL algorithm. Table 2 shows the hyperparameters that are shared across all methods.Table 3 and Table 4 show the algorithm-specific hyperparameters for C-TeC and the baselines, respectively.

| Hyperparameter | Value |
|---|---|
| num_timesteps | 500,000,000 |
| max_replay_size | 10,000 |
| min_replay_size | 1,000 |
| episode_length | 1,000 |
| discounting | 0.99 |
| num_envs | 1024 (256 for humanoid_u_maze) |
| batch_size | 1024 (256 for humanoid_u_maze) |
| multiplier_num_sgd_steps | 1 |
| action_repeat | 1 |
| unroll_length | 62 |
| policy_lr | 3e-4 |
| critic_lr | 3e-4 |
| hidden layers (for both actor and critic) | [256,256] |

Table 2: Hyperparameters for all methods in robotics environments

| Hyperparameter | Value |
|---|---|
| contrastive_lr | 3e-4 |
| contrastive_loss_function | InfoNCE |
| similarity_function | L1 |
| logsumexp_penalty | 0.1 |
| hidden layers (for both encoders) | [1024,1024] |
| representation dimension | 64 |

Table 3: Hyperparameters for C-TeC in robotics environments

| Hyperparameter | Value |
|---|---|
| rnd encoder lr | 3e-4 |
| rnd embedding dim | 512 |
| rnd encoder hidden layers | [256, 256] |
| icm encoder lr (forward and inverse models) | 3e-4 |
| icm embeddings_dim | 512 |
| icm encoders hidden layers | [1024, 1024] |
| icm weight on forward loss | 0.2 |
| apt contrastive lr | 3e-4 |
| apt similarity function | L1 |
| apt contrastive hidden layers | [1024, 1024] |
| apt representation dimension | 64 |
| Augmentation type | $\mathcal{N}(0, 0.5)$ |

Table 4: Hyperparameters for baselines in robotics environments

### E.2 CRAFTAX

In Craftax, we used PPO as the RL algorithm[3]. Table 5 shows the hyperparameters shared across all methods. Table 6 and Table 7 show the algorithm-specific hyperparameters for C-TeC and the baselines, respectively.

---

[3]https://github.com/MichaelTMatthews/Craftax_Baselines

| Hyperparameter | Value |
|---|---|
| num_timesteps | 1,000,000,000 |
| num_steps | 64 |
| learning_rate | 2e-4 |
| anneal_learning_rate | True |
| update_epochs | 4 |
| discounting | 0.99 |
| gae_lambda | 0.8 |
| clip_epsilon | 0.2 |
| ent_coef | 0.01 |
| max_grad_norm | 1.0 |
| activation | tanh |
| action_repeat | 1 |
| RNN_layers (GRU) | [512 (embedding dim),512 (hidden dim)] |
| hidden layers (both actor and value) | [512, 512] |

Table 5: Hyperparameters for all methods in robotics environments

| Hyperparameter | Value |
|---|---|
| contrastive_lr | 3e-4 |
| contrastive_loss_function | InfoNCE |
| similarity_function | L2 |
| Discounting | 0.3 |
| logsumexp_penalty | 0.0 |
| hidden layers (for both encoders) | [1024,1024,1024] |
| representation dimension | 64 |

Table 6: Hyperparameters for C-TeC in Craftax

| Hyperparameter | Value |
|---|---|
| rnd encoder lr | 3e-4 |
| rnd embedding dim | 512 |
| rnd encoder hidden layers | [256, 256] |
| icm encoder lr (forward and inverse models) | 3e-4 |
| icm embeddings_dim | 512 |
| icm encoders hidden layers | [256, 256] |
| icm weight on forward loss | 1.0 |
| e3b (icm) lambda | 0.1 |

Table 7: Hyperparameters for baselines in Craftax

### E.3 ENVIRONMENT DETAILS

- **Ant-hardest-maze** The observation space of this environment has 29 dimensions, consisting of joint angles, angular velocities, and the x,y position of the ant's torso. The action space is 7-dimensional, representing the torque applied to each joint.

- **Humanoid-u-maze** The observation space of this environment has 268 dimensions, consisting of joint angles, angular velocities, and the x,y position of the humanoid's torso. The action space is 17-dimensional, representing the torque applied to each joint.

- **Arm-binpick-hard** The observation space of this environment has 18 dimensions, consisting of joint angles, angular velocities, the cube position, and the end-effector position and offset. The action space is 5-dimensional, representing the displacement of the end-effector.

- **Craftax-Classic** The observation space is a one-hot encoding of size 1345, capturing player information (inventory, health, hunger, attributes, etc.) as well as the types of blocks and creatures within the player's visual field. The action space is discrete and consists of 17 actions.

### E.4 DETAILS ON C-TEC REWARD

One important detail is that the policy's objective is slightly different from the (negative) representation objective ( Equation (2)) because it omits the log-sum-exp term. This can be seen by rewriting the reward function as follows:

$$r_{\text{intr}}(s,a) = \mathbb{E}_{p_{\mathcal{T}}(s_f|s,a)}[\underbrace{\|\phi(s,a) - \psi(s_f)\| + \log\sum_{s'_f} e^{-\|\phi(s,a)-\psi(s'_f)\|}}_{\text{(neg) contrastive loss}} - \log\sum_{s'_f} e^{-\|\phi(s,a)-\psi(s'_f)\|}].$$

$$(6)$$

To further gain intuition for what this is doing, we note that (in practice) the $\phi(s,a)$ representations are quite similar to the $\psi(s)$ representation evaluated at the same state. Thus, we can approximate this second term as

$$\log\sum_{s'_f} e^{-\|\psi(s)-\psi(s'_f)\|} \approx \log\hat{p}(s), \tag{7}$$

which we identify as a kernel density estimate of the marginal likelihood of state $s$ under the replay buffer distribution $p_{\mathcal{T}}(s)$. This observation helps explain why including the log-sum-exp term in the reward would degrade performance – it effectively corresponds to *minimizing* state entropy, which can often hinder exploration, especially in environments without much noise (Zheng et al., 2025). One additional consideration here is that, because the likelihood is measured using learned representations, it is sensitive to the policy's understanding of the environment. While ordinarily maximizing state entropy can lead to degenerate solutions (like the noisy TV), our approach mitigates this problem because the contrastive representations will only learn features that are predictive of future states (hence, they would ignore a noisy TV).

#### E.4.1 VARIANCE REDUCTION IN THE REWARD ESTIMATE

We can decrease the variance in our estimate of the expectation in Equation (3) by looking at all future states $s_f = s_{t+1}, s_{t+2}, \cdots$ and weighting each summand by $\gamma^i$:

$$r_t = \mathbb{E}_{p(s_f|s_t,a_t)}[r_{\text{int}}(s_t, a_t, s_f)] \tag{8}$$

$$= \mathbb{E}_{p(s_f|s_t,a_t)}[\|\phi(s,a) - \psi(s_{t'})\|_2] \tag{9}$$

$$\approx \frac{1 - \gamma^{H-t}}{1 - \gamma} \sum_{t'=t}^{H} \gamma^{t'-t}\|\phi(s,a) - \psi(s_{t'})\|_2 \tag{10}$$

The (unbiased) approximation comes because we only look at future states that occur in one trajectory, and other trajectories might visit different future states. The ugly fraction is the normalizing constant for a truncated geometric series. In the last line, note that the summation $\sum_{t'=t}^{H} \gamma^{t'-t}\psi(s_{t'})$ can be quickly computed for every $r_t$ by starting at $T = H$ and decrementing $t$, updating $\psi_{\text{sum}} = \psi(s_t) + \gamma\psi_{\text{sum}}$. This is the same trick that's usually used for computing the empirical future returns in REINFORCE, and decreases compute from $\mathcal{O}(H^2)$ to $\mathcal{O}(H)$. We use this estimator in Craftax-Classic but we found that omitting the normalization term results in much better performance.

## F COMPUTE RESOURCES

In all experiments, we use 2 CPUs, a single GPU, and 8 GB of RAM. The specific GPU type varies depending on the job scheduling system, but most experiments run on NVIDIA RTX 8000 or V100 GPUs. Training in the robotics environments takes approximately 24 hours on average, while Craftax experiments require around 30 hours.

## G ABLATION STUDY

To understand the contribution of each component to the overall performance of C-TeC, we conduct an ablation study on several key elements of the algorithm, illustrated in the following section.

### G.1 REPRESENTATION NORMALIZATION

Is it important to normalize the contrastive representations when computing the intrinsic reward? To answer this question, we compare the exploration performance of C-TeC across all environments, keeping all hyperparameters fixed except for the normalization of the representations.

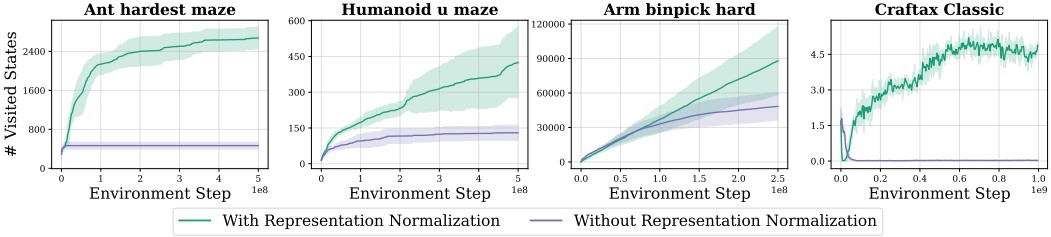

Figure 11: **Normalizing the contrastive representations.** Normalizing the representations is crucial for effective exploration—using unnormalized representations significantly degrades exploration performance.

### G.2 CONTRASTIVE LOSSES

We compare the performance of C-TeC using different contrastive loss functions. Specifically, we evaluate InfoNCE, symmetric InfoNCE, NCE (Hjelm et al., 2019), FlatNCE (Chen et al., 2021), and a Monte-Carlo version of the forward-backward (FB) (Touati & Ollivier, 2021) loss, as defined in [Equation (11)–Equation (15)]. Figure 12 presents the results. Overall, NCE leads to poorer exploration, particularly in Craftax. InfoNCE and symmetric InfoNCE exhibit similar performance across all environments. In general, the method is reasonably robust to the choice of contrastive loss.

$$\mathcal{L}_{\text{InfoNCE}}(\theta) = -\sum_{i=1}^{K} \log \left( \frac{e^{C_\theta((s_i,a_i),s_f^{(i)})}}{\sum_{j=1}^{K} e^{C_\theta((s_i,a_i),s_f^{(j)})}} \right) \tag{11}$$

$$\mathcal{L}_{\text{symmetric\_InfoNCE}}(\theta) = -\left[ \sum_{i=1}^{K} \log \left( \frac{e^{C_\theta((s_i,a_i),s_f^{(i)})}}{\sum_{j=1}^{K} e^{C_\theta((s_i,a_i),s_f^{(j)})}} \right) + \log \left( \frac{e^{C_\theta((s_i,a_i),s_f^{(i)})}}{\sum_{j=1}^{K} e^{C_\theta((s_j,a_j),s_f^{(i)})}} \right) \right] \tag{12}$$

$$\mathcal{L}_{\text{Binary(NCE)}}(\theta) = -\left[ \sum_{i=1}^{K} \log \left( \sigma \left( C_\theta((s_i,a_i), s_f^{(i)}) \right) \right) - \sum_{j=2}^{K} \log \left( 1 - \sigma \left( C_\theta((s_i,a_i), s_f^{(j)}) \right) \right) \right] \tag{13}$$

$$\mathcal{L}_{\text{FlatNCE}}(\theta) = -\sum_{i=1}^{K} \log \left( \frac{\sum_{j=1}^{K} e^{C_\theta(s_i,a_i,s_f^{(j)}) - C_\theta(s_i,a_i,s_f^{(i)})}}{\text{detach} \left[ \sum_{j=1}^{K} e^{C_\theta(s_i,a_i,s_f^{(j)}) - C_\theta(s_i,a_i,s_f^{(i)})} \right]} \right) \tag{14}$$

$$\mathcal{L}_{\text{FB}}(\theta) = -\sum_{i=1}^{K} \left( e^{C_\theta(s_i,a_i,s_f^{(i)})} \right) + \frac{1}{2(K-1)} \sum_{i=1}^{K} \sum_{\substack{j=1 \\ j \neq i}}^{K} \left( e^{C_\theta(s_i,a_i,s_f^{(j)})} \right)^2 \tag{15}$$

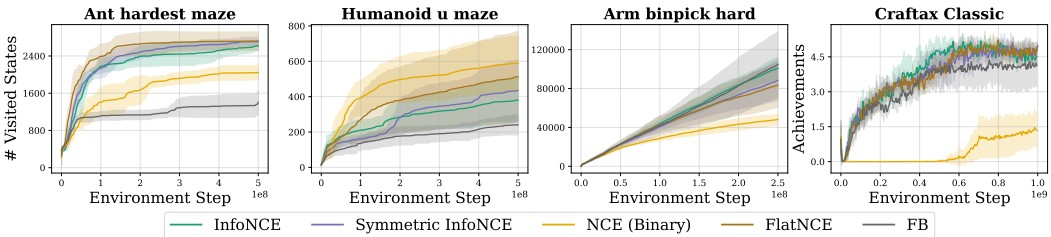

Figure 12: **Comparison of Different Contrastive Losses.** Overall, C-TeC is robust to the choice of contrastive loss. A notable exception is the Binary NCE loss in Craftax, where it performs relatively poorly.

### G.3 CONTRASTIVE CRITIC FUNCTIONS

We compare four critic similarity functions shown below:

$$C_\theta((s_t, a_t), s_f)_{L1} = -||\phi_\theta(s_t, a_t) - \psi_\theta(s_f)||_1. \tag{16}$$

$$C_\theta((s_t, a_t), s_f)_{L2} = -||\phi_\theta(s_t, a_t) - \psi_\theta(s_f)||_2 \tag{17}$$

$$C_\theta((s_t, a_t), s_f)_{L2-w/o-sqrt} = -||\phi_\theta(s_t, a_t) - \psi_\theta(s_f)||_2^2 \tag{18}$$

$$C_\theta((s_t, a_t), s_f)_{dot} = -\phi_\theta(s_t, a_t)^\top \psi_\theta(s_f) \tag{19}$$

Figure 13 shows the results. In general, using the $L_1$ distance yields the best performance across the robotic environments, while $L_2$ performs better in Craftax. This highlights the importance of this design choice and suggests that some tuning may be required to select the most effective critic function.

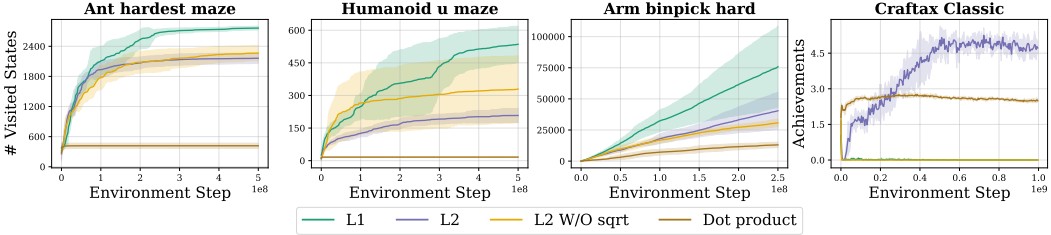

Figure 13: **Comparison of Critic function.** Overall, the $L_1$ distance yields the best performance across the robotic environments, while $L_2$ performs better in Craftax.

### G.4 CONTRASTIVE CRITIC ARCHITECTURE

In this ablation we compare two architectures of the contrastive critic, the separable architecture $(\phi_\theta(s_t, a_t), \psi_\theta(s_f))$, which is the one we use in all of our experiment, and the monolithic critic $f_\theta$ i.e., a single model that takes in triplet $f(s, a, s_f)$.

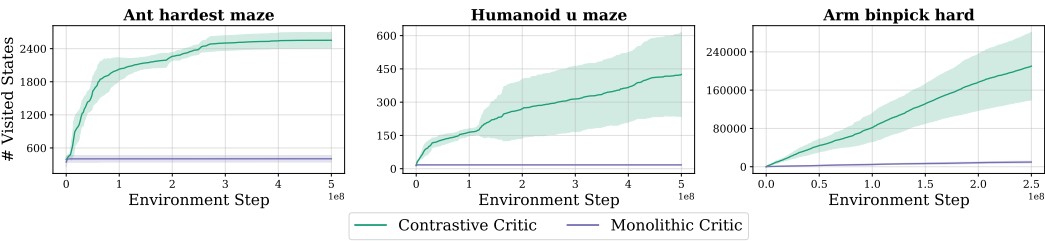

Figure 14: **Critic parameterization** Using a monolithic critic results in poor exploration performance, while using the separable architecture results in much better exploration. This shows the importance of the critic parameterization as a distance function between two representations.

Importantly, these experiments show that the factorized representation parameterization is a necessary (relative to the monolithic critic) condition for effective exploration. We discuss the possible failure mode of using the monolithic critic in Section 5.2. These experiments do not demonstrate sufficiency, and we claim that the information-theoretic interpretation for a critic that fully captures the point-wise MI is still useful for analysis.

## G.5 Forward vs. reverse KL

As mentioned in Section 5.1, we hypothesized that the reverse KL C-TeC reward is important for exploration. As it encourages mode-seeking behavior (prioritizing unfamiliar states), to test this hypothesis we run C-TeC with the negative-forward KL reward (Equation (20)), the results shown in Appendix G.5 indicates that using the reverse is necessary for exploration.

$$\mathbb{E}[r_{\text{intr}}(s_t, a_t)] = -\mathbb{E}_{p_{\mathcal{T}}(s_f)}\left[\log \frac{p_{\mathcal{T}}(s_f)}{p_{\mathcal{T}}(s_f \mid s_t, a_t)}\right] \tag{20}$$
$$= -D_{\text{KL}}[p_{\mathcal{T}}(s_f) \,\|\, p_{\mathcal{T}}(s_f \mid s_t, a_t))] \leq 0. \qquad (D_{\text{KL}} \text{ is always non-negative.})$$

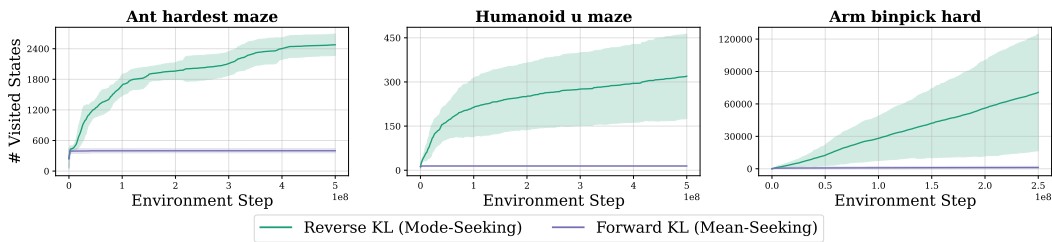

Figure 15: **Forward vs Reverse KL** C-TeC with the reverse KL reward promotes mode-seeking behavior which encourages the agent to prioritize visiting unfamiliar states resulting in much better exploration.

## G.6 Future state sampling strategy

In this experiment (Figure 16), we investigate the sensitivity of C-TeC to the future state sampling strategy. Specifically, we consider two variants in addition to the geometric sampling. The first is uniformly sampling from the future. Unlike geometric sampling, uniform sampling does not prefer states that are sooner in the future over later ones. The second is geometric sampling with an increasing $\gamma$ value. The intuition behind this strategy is that exploring nearby states is easier for the agent at the start of training, and as the agent becomes better at exploring them, it can progressively explore farther states in the future. We refer to this strategy as the $\gamma$-schedule, and we experiment with two different starting values of $\gamma$: one ranging from $\gamma = 0.9$ to $\gamma = 0.99$, and another from $\gamma = 0.1$ to $\gamma = 0.99$. The results are shown in Figure 16. Regardless of the future state sampling strategy, the contrastive method explores better than the baselines in all three environments and appears robust.

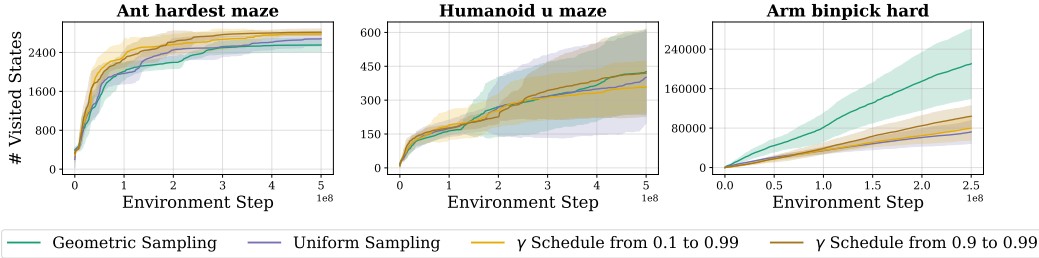

Figure 16: **Sensitivity to future state sampling strategy.** We compare variants of C-TeC with different future state sampling strategies, the method is robust to the choice of the sampling strategy and all the variants outperform the baselines.

## G.7 EFFECT OF THE CONTRASTIVE MODEL DISCOUNT FACTOR

We study the effect of the discount factor $\gamma_{cl}$ in Equation (1), which defines the sampling window of future states; this discount is distinct from the discount factor $\gamma$ used in the underlying RL algorithm. We found that, in general, a discount value of $\gamma_{cl} = 0.99$ yields good exploration performance; however, we suspect that adjusting the discount might result in better performance depending on the environment. Figure 17 shows the results in the robotic environments. In humanoid-u-maze, smaller values of $\gamma_{cl}$ result in better performance. In ant-hardest-maze, a discount of $\gamma_{cl} = 0.9$ leads to faster exploration; however, by the end of training, performance is similar. Finally, in arm-binpick-hard, a discount of $\gamma_{cl} = 0.99$ achieves the best performance.

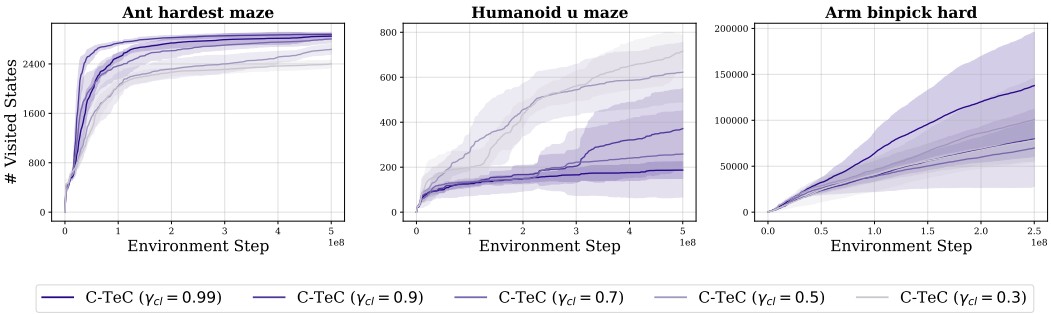

Figure 17: **Sensitivity of C-TeC to the discount factor** Different environments require different discount factors to obtain the best exploration performance.

## G.8 EFFECT OF THE TEMPERATURE PARAMETER $\tau$

In practice, $\tau$ (Equation (2)) is learned during training as an additional learnable parameter. However, we study the effect of different fixed values of $\tau$. Intuitively, we can think of $\tau$ as a weight on the alignment and uniformity terms in the contrastive loss (Wang & Isola, 2020): smaller values of $\tau$ put more weight on the alignment term, while larger values put more weight on the uniformity term. Figure 18 illustrates that a temperature value of 1 often results in good performance, indicating the importance of both alignment and uniformity in learning representations for exploration.

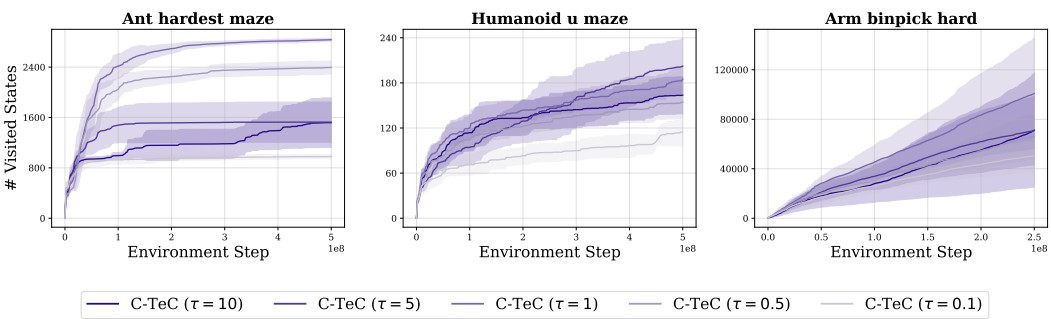

Figure 18: **Sensitivity of C-TeC to the temperature $\tau$ in the contrastive loss (Equation (2))** C-TeC is sensitive to the temperature parameter, however, in general, a value of $\tau = 1$ yields a large state coverage across environments.

## G.9 IN-EPISODE VS ACROSS-EPISODE NEGATIVE SAMPLING

Contrastive learning requires negative sampling to prevent representations from collapsing. In practice, for each batch sample, the positive examples of other samples are treated as negatives, following common practice (Chen et al., 2020). However, it might be desirable to sample negatives from the same trajectory, and results from Ziarko et al. (2025) suggest that doing so leads to learning better temporal structures compared to standard practice. To sample negatives in-episode, we utilize the

method from Ziarko et al. (2025), where we control the amount of in-episode negatives by duplicating the same trajectory in the sampled batch. We adjust the amount of duplication using a repetition factor, where a repetition factor of 1 is equivalent to sampling negatives across episodes, and larger values indicate more in-episode samples. Figure 19 shows that in-episode sampling can result in better exploration performance, particularly in `humanoid-u-maze` and `arm-binpick-hard`.

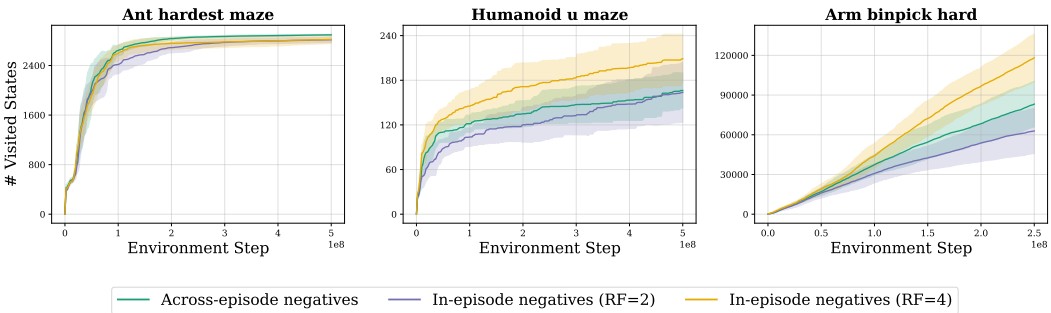

Figure 19: **In-episode vs Across-episode negative sampling** Sampling in-episode negatives results in additional performance boost in `humanoid-u-maze` and `arm-binpick-hard`.

## H    EXPLORATION IN NOISY TV SETTING

We investigate C-TeC performance in the presence of a noisy TV state, we run this experiment on a modified grid environment from xland-minigrid (Nikulin et al., 2024) of size $256 \times 256$ Figure 21 with a noisy TV region. We did not observe any evidence of worse exploration performance namely the agent has covered all the states in the grid world, Appendix H shows the state coverage of C-TeC compared to the maximum coverage.

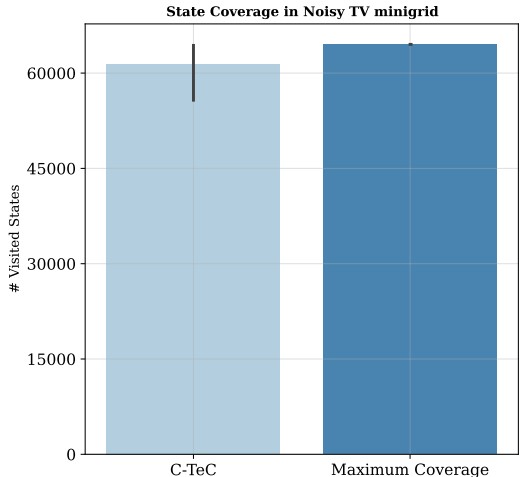

Figure 20: **C-TeC Coverage in noisy TV setting** C-TeC can effectively explore in the presence of noisy states

## I    COMPARISON WITH PREVIOUS METHODS

At a high level, C-TeC is related to other intrinsic exploration objectives that reward uncertainty. Objectives such as RND (Burda et al., 2019b) and Disagreement (Pathak et al., 2019) explore unfamiliar states, presumably leading to these states becoming more familiar in future rounds. A related method, CURL (Du et al., 2021), also relies on using a negative contrastive similarity score

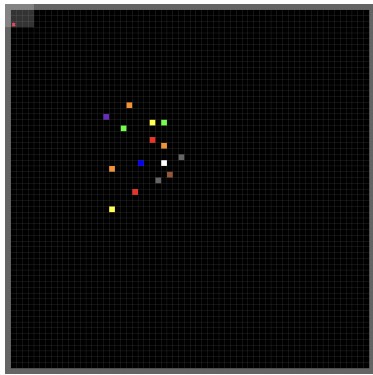

Figure 21: Xland-Minigrid (Nikulin et al., 2024) with noisy TV states indicated by the random colors.

for exploration like C-TeC. CURL prioritizes exploration over states with high error/low similarity scores with augmented states; however, the contrastive features learned in CURL are not temporal and can be concretely related to prediction error.

The key difference between prior methods and C-TeC lies in the usage of temporal contrastive features. Our method drives the agent to explore areas where *future* outcomes have been seen but appear improbable. Taken together, our analysis and results show that temporal contrastive representations are simple yet powerful frameworks for intrinsic motivation.

## J  INTRINSIC REWARD INTERPRETATION

**On information-theoretic interpretation of reward.** The intrinsic reward with representations rewards $(s, a)$ pairs that result in the largest additional number of bits needed to encode the representation induced $p_{\phi,\psi}(s_f \mid s_t, a_t)$ with a code optimized for the marginal $p_{\phi,\psi}(s_f)$. In other words, it prioritizes exploration in areas where the representation encoding schemes is highly inefficient.

**On information-theoretic interpretation of objective assuming perfect estimation of point-wise MI.** We assume that representations perfectly capture point-wise MI. Taking an additional expectation of the roll-out state-occupancy reveals that the PPO/SAC objective is a minimization of MI

$$J^\pi = \mathbb{E}_{p_\pi(s,a), p_\mathcal{T}(s_f|s,a)}[r_\text{intr}(s_t, a_t)] \approx -I[S_f; S_\pi, A_\pi] \tag{21}$$

where $p_\pi$ is the policy induced discounted state-occupancy measure (see Eq. 1).

**On C-TeC as a Two-Player Game**

In addition to quantifying temporal similarity, the converged InfoNCE loss $\mathcal{L}^*_\text{CRL}$ provides a lower bound on the mutual information (MI) (Oord et al., 2018; Eysenbach et al., 2021):

$$I(S_f; S_t, A_t) \geq \log K - \mathcal{L}^*_\text{CRL}(\mathcal{B}; \theta).$$

Contrastive learning finds representations that maximize a lower bound on the MI between *current* states and actions and *future* state distributions.

Thus, we can view C-TeC as a two-player game over an ever-expanding buffer. Namely, the CL step learns to minimize $\mathcal{L}_\text{CRL}$. Meanwhile, the policy objective learns to approximately maximize $\mathcal{L}_\text{CRL}$ when state-action pairs are strictly drawn from the roll-out policy (as opposed to the entire buffer), and the conditional and marginal future-state distributions are still defined over the buffer.

### J.1  NO (ACHIEVABLE) TRIVIAL FIXED POINTS

Does C-TeC have stable fixed points? Without additional simplifications, this problem is intractable. Notably, standard analysis would fail to prove convergence due to the non-convexity/concavity of the objectives. While the zero-gradient condition for the InfoNCE objective is clear, the zero-gradient condition for the objective is not obvious due to the complex relationship between $\pi$ and the state occupancy measure.

A more aggressive simplification that can simplify analysis of the global optimum is to (1) assume that the policy optimization is done directly over $S_\pi$ and $A_\pi$ and (2) assume the representations perfectly capture the point-wise MI. Furthermore, we assume that future states are exclusively sampled from the one-step transition dynamics and are deterministic.

In practice, these assumptions are very unrealistic; however, such simplifications have been used in prior work on unsupervised RL to give a conceptual picture of exploration methods (Pitis et al., 2020). Throughout, we assume fully expressive representations that capture the point-wise MI – thus, we are strictly analyzing fixed points and fixed point-stability/achievability without taking into account representations.

Though the following analysis assumes a discrete setting (summations vs. integrals, Kronecker Deltas vs. Dirac Deltas), we do not directly invoke the assumption of discreteness. The conclusions should continue to hold in the continuous case assuming all relevant probability distributions are bounded and smooth.

With these simplifications, the InfoNCE objective reduces to:

$$\max_{\phi,\psi} \ [\log K - \mathcal{L}_{\phi,\psi}(Z_\mathcal{T}, F_\mathcal{T})] \xrightarrow[K\to\infty, \text{infinitely expressive reps}]{} I\big(S_\mathcal{T}, A_\mathcal{T}; S_f\big).$$

Because the "policy" optimization is fixed in $p(s_f \mid s, a)$, the MI $I(S_f; S_\pi, A_\pi)$ (see Eq. 21) is concave in $p_\pi(s, a)$ and $p_\pi(s_f)$ (Cover & Thomas, 1991). Our objective has now reduced to a constrained optimization problem with conditions $\sum_{s,a} p_\pi(s, a) = 1$ and $p_\pi(s, a) \geq 0$ for all $(s, a) \in \mathcal{S} \times \mathcal{A}$.

Consider the fixed point conditions given by the Lagrangian that is Lipschitz-continuous over the probability simplex $\Delta_\mathcal{S}$. Let $\lambda$ and $\mu(s, a)$ denote the Lagrange multipliers for the normalization and non-negativity conditions respectively. Then, the full Lagrangian $\mathcal{L}_{\text{Lagrangian}}$ is as follows:

$$\mathcal{L}_{\text{Lagrangian}}(p_\pi, \lambda, \mu) = I(S_\pi, A_\pi; S_f) + \lambda\Big(\sum_{s,a} p_\pi(s, a) - 1\Big) - \sum_{s,a} \mu(s, a) p_\pi(s, a).$$

Note that by complementary slackness, we have $\mu(s, a) p(s, a) = 0$. Taking the functional derivative of $\mathcal{L}_{\text{Lagrangian}}$ with respect to distribution $p(s, a)$ yields the KL-divergence:

$$\frac{\delta \mathcal{L}_{\text{Lagrangian}}}{\delta p_\pi}[s, a] = D_{KL}[p_\mathcal{T}(s_f \mid s, a) || p_\mathcal{T}(s_f)] - 1 + \lambda - \mu(s, a).$$

By the complementary slackness, the distribution $p_\pi(s, a)$ is a fixed point if the KL-divergence $D_{KL}[p_\mathcal{T}(s_f \mid s, a) || p_\mathcal{T}(s_f)]$ is *constant* for any $(s, a)$ where $p_\pi(s, a)$ has support. Any deviation would lead to a non-zero gradient at the point $(s, a)$. In other words, all conditional trajectory future state distributions look equally "far" from the marginal.

Stationarity over *iterations* of C-TeC requires an additional condition: that the $D_{KL}$ remains constant over all $(s, a)$ *after* updating buffer $p_\mathcal{T}(s_f)$ with states encountered during the roll-out. A model of this is reweighing the marginal with the rollout probability distribution $p_\pi(s_f) = \sum_{s,a} p_\pi(s_f \mid s, a) p_\pi(s, a)$:

$$D_{KL}[p_\mathcal{T}(s_f \mid s, a) || p'(s_f)] = D_{KL}[p_\mathcal{T}(s_f \mid s, a) || (1 - \alpha) \cdot p_\mathcal{T}(s_f) + \alpha \cdot p_\pi(s_f)] \quad (22)$$

where $0 < \alpha < 1$. Again, we assume deterministic dynamics for simplicity and that $s_f$ is always the next state (i.e. small discount factor like the Craftax setting) so the conditional distribution does not change; otherwise, we have no easy way of determining the change in $p_\mathcal{T}(s_f \mid s, a)$ after roll-out.

We drop subscripts on transitions and simplify:

$$LHS = \sum_{s_f} p(s_f \mid s, a) \log \frac{p(s_f \mid s, a)}{p_\mathcal{T}(s_f)} - \sum_{s_f} p(s_f \mid s, a) \log \frac{p_\mathcal{T}(s_f)}{(1 - \alpha) \cdot p_\mathcal{T}(s_f) + \alpha \cdot p_\pi(s_f)}$$

$$= D_{KL}[p_\mathcal{T}(s_f \mid s, a) || p_\mathcal{T}(s_f)] - \mathbb{E}_{p(s_f|s,a)}\Big[\log \frac{p_\mathcal{T}(s_f)}{(1 - \alpha) \cdot p_\mathcal{T}(s_f) + \alpha \cdot p_\pi(s_f)}\Big]$$

$$= C_{\text{old}} - \mathbb{E}_{p(s_f|s,a)}\Big[\log \frac{p_\mathcal{T}(s_f)}{(1 - \alpha) \cdot p_\mathcal{T}(s_f) + \alpha \cdot p_\pi(s_f)}\Big]$$

where $C_{\text{old}}$ is the old constant $D_{KL}$ across $(s, a)$. Thus, for the LHS to also be constant across $(s, a)$, the difference must also be constant. We assume that transitions are nontrivial (as in, $p(s_f \mid s, a) \neq p(s_f)$). This implies that the updated $D_{KL}$ remains constant iff

$$(1 - \alpha) \cdot p_{\mathcal{T}}(s_f) + \alpha \cdot p_\pi(s_f) = p_{\mathcal{T}}(s_f)$$
$$\Rightarrow p_{\mathcal{T}}(s_f) = p_\pi(s_f).$$

Under the assumptions of one-step, deterministic transitions and the $\alpha$-reweighing of the buffer distribution, the distribution $p_\pi(s, a)$ remains a fixed point iff the roll-out future distribution and buffer future distribution are identical.

What is the stability of these fixed points? We can do linear fixed-point stability analysis by calculating the Jacobian of the update, where prime (') denotes the next-step $\delta p_\pi(s_f)$. The update of $\delta p_\pi(s_f)$ is as follows:

$$\delta p'_\pi(s_f) = \delta p_\pi(s_f) - \eta p(s_f \mid s, a) \Big[\Big(\nabla^2_{p_\pi(s,a)} I(S_\pi, A_\pi; S_f)\Big) \delta p_\pi\Big](s, a) \tag{23}$$

$$= \delta p_\pi(s_f) - \eta \Big[\Big(\nabla^2_{p_\pi(s_f)} I(S_\pi, A_\pi; S_f)\Big) \delta p_\pi\Big](s_f) \qquad \text{(change of vars.)}$$

$$= \Big(I - \eta \nabla^2_{p_\pi(s_f)} I(S_\pi, A_\pi; S_f)\Big) \delta p_\pi(s_f), \tag{24}$$

We can similarly calculate the update for $\delta p_{\mathcal{T}}(s_f)$:

$$\delta p'_{\mathcal{T}}(s_f) = \alpha \Big(I - \eta \nabla^2_{p_\pi(s_f)} I(S_\pi, A_\pi; S_f)\Big) \delta p_\pi(s_f) \qquad \text{(weight new traj.)}$$

$$+ (1 - \alpha) \delta p_{\mathcal{T}}(s_f). \qquad \text{(down-weight old traj.)}$$

Thus, the equation relating $(\delta p_\pi(s_f), \delta p_{\mathcal{T}}(s_f))$ and $(\delta p'_\pi(s_f), \delta p'_{\mathcal{T}}(s_f))$ is

$$\begin{pmatrix} \delta p'_\pi(s_f) \\ \delta p'_{\mathcal{T}}(s_f) \end{pmatrix} = \underbrace{\begin{pmatrix} I - \eta H & 0 \\ \alpha (I - \eta H) & (1 - \alpha) I \end{pmatrix}}_{J} \begin{pmatrix} \delta p_\pi(s_f) \\ \delta p_{\mathcal{T}}(s_f) \end{pmatrix}.$$

to first order in iteration time $\tau$, where $H$ is the Hessian of the MI with respect to $p_\pi(s_f)$. Because the MI is concave in $p_\pi(s_f)$, the Hessian $H$ is negative semi-definite; note that if $H$ has any negative eigenvalues at the fixed point, the Jacobian would have at least one eigenvalue $> 1$. Thus, the non-vertex fixed points in the product of two probability simplices $\Delta_{\mathcal{S}} \times \Delta_{\mathcal{S}}$ (where $p_{\mathcal{T}} = p_\pi(s_f)$) are either unstable, where at least one direction corresponds to an eigenvalue $> 1$ in the Jacobian, or semi-stable fixed points, where the MI is locally flat at the fixed point. Finally, fixed points at the vertices of the probability simplex (Delta functions) are uninteresting and are not observed in practice.

For an arbitrary MDP, we note that semi-stable fixed points are generally hard to achieve: a nontrivial, non-constant transition function, random roll-outs at initialization, the mixture of policies in the buffer, and newly encountered states prevent such semi-stable states from being easily accessible. Particularly, the random roll-outs help prevent no-op from being a trivial fixed point.

This analysis shows that there are no easily-obtainable, stable fixed points for standard MDPs even under aggressive simplifications, implying constantly evolving probability distributions. Future work remains to investigate the existence of dynamical steady-states and whether the reached probability distributions cover a large region of the probability simplex.

## J.2 DIDACTIC TOY EXAMPLE

We present a didactic example that clarifies the mode-seeking interpretation of C-TeC reward and the difference between our reward and the ETD reward by Jiang et al. (2025). The ETD reward (at convergence of contrastive representations) can be written as follows:

$$r_{\text{ETD}} = \min_{k \in [0,t)} \log \frac{p(s_t \mid s_t)}{p(s_t \mid s_{k<t})}. \tag{25}$$

ETD rewards states $s_t$ that are improbable from prior episodic states $s_k$ (low $p(s_+ = s_t \mid s = s_k)$) relative to $p(s_+ = s_t \mid s_+ = s_t)$, where the reward is computed in the worst case over the episodic

memory. The overall ETD objective is to maximize the discounted sum of worst-case temporal distances. The C-TeC reward can be expressed as

$$\mathbb{E}[r_{\text{C-TeC}}] = \mathbb{E}\left[\log \frac{p(s_f)}{p(s_f \mid s, a)}\right]. \tag{26}$$

C-TeC rewards states that are that are improbable (have low $p(s_f \mid s, a)$) relative to the overall marginal $p(s_f)$. Thus, C-TeC rewards states present in the buffer ("familiarity") that are tough to reach from the current state-action.

We show the difference in optimal agent behavior from maximizing Equation (25) and Equation (26) in a simple MDP (Figure 22). The MDP consists of a root node connected to a right and a left branch. All trajectories begin from this root node. The left branch contains fast dynamics: the agent deterministically moves down the branch to the leaf node, progressing one level per timestep.

The right branch contains sticky dynamics. With $90\%$ probability, the agent will remain stuck at the state for a given timestep. With $10\%$ probability, the agent will progress down the tree. The dynamics of the agent are independent of the policy after choosing the branch. Thus, this problem is a 2-armed bandit where the agent chooses a left or right branch. We consider episodes of length 30 with $\gamma = 0.99$ for both the discount factor and future state sampling in C-TeC.

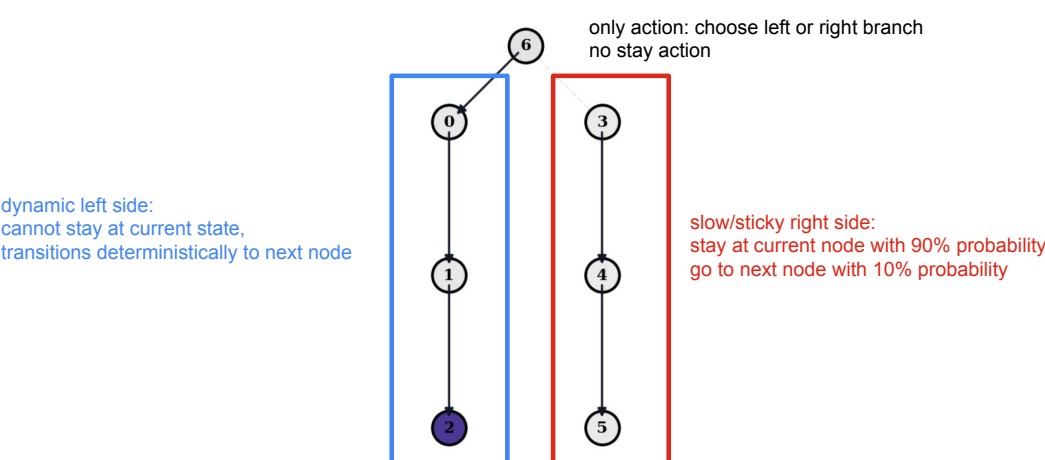

Figure 22: From the root, trajectories go either left or right. The left branch has fast, deterministic dynamics: the agent moves one level per timestep to the leaf. The right branch has slow, sticky dynamics: the agent stays in place with some probability and progresses with the remaining probability. After choosing a branch, the dynamics are policy-independent, reducing the problem to a two-armed bandit.

An agent that maximizes the C-TeC reward is incentivized to match the marginal $p(s_f)$ and $p(s_f \mid s, a)$. We visualize the future state distributions for different nodes in Figure 23. Clearly, the left side of the MDP leads to $p(s_f \mid s, a)$ that much more readily matches the marginal $p(s_f)$: the state visitation has a mode at leaf node 2 with very little probability mass on nodes 0 and 1 in the buffer. Meanwhile, the right branch assigns more visitation probability to nodes 3 and 4 from the slow dynamics: the distributions $p(s_f \mid s, a)$ are less aligned with the marginal. Correspondingly, the CTEC agent chooses the left, fast-moving branch, reflecting our mode-seeking interpretation, and the ETD agent chooses the right, slow-moving/sticky branch where nodes are temporally distant.

To visualize the mode-seeking nature of C-TeC, Figure 24 shows the discounted future state distribution for C-TeC on the preferred (left) side of the MDP and ETD on the preferred (right) side of the MDP. Even though the graphical MDP structure on the left and right are identical, the difference in the transition kernel splits the methods' behaviors. The left side of the MDP more readily enables $p(s_+ \mid s)$ to seek modes in $p(s_+)$

**Learned Policies (Arrows) and Last 1000 Episode State Visitations (Normalized)**

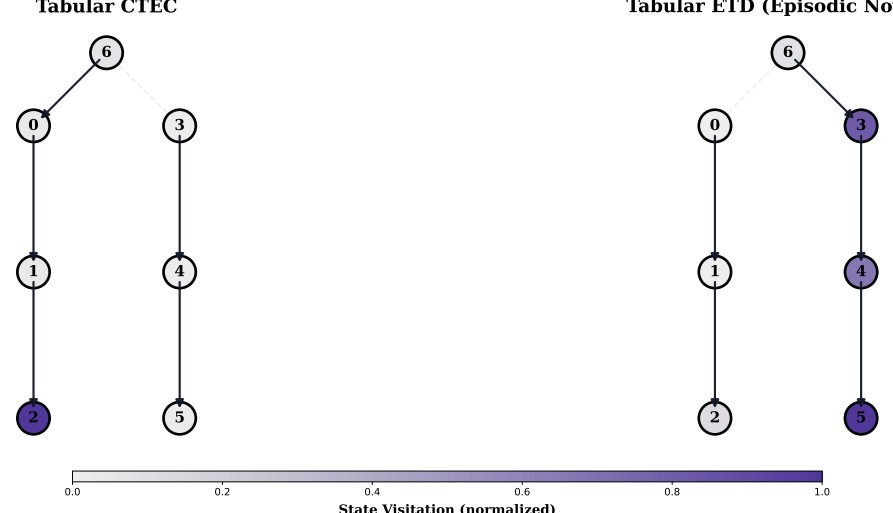

Figure 23: State visitation of C-TeC and ETD, C-TeC distribution has a mode on the most left node, while ETD prefers to stay in any state in the right branch

**Discounted Future State Distributions**

Figure 24: Future state distribution in the toy MDP in Figure 22. C-TeC prefers the states in the left branch with fast, deterministic dynamics while ETD prefers the right branch with slow, sticky dynamics.

## K  FORWARD-LOOKING VS BACKWARD-LOOKING REWARDS

In this section, we illustrate when a forward-looking reward like C-TeC may lead to divergent behavior when compared with ETD-type backward-looking reward (Jiang et al., 2025). We work in the tabular setting to isolate our results from any function approximation error.

We claim that forward-looking rewards may be beneficial when there are environment transitions with arbitrarily long waiting times. Consider an MDP in which a subgraph is a tree with actions $\mathcal{A} = \{\text{left child}, \text{right child}, \text{stay}\}$. Actions are fully deterministic from the root, with the exception of a 10% chance of a random action at any node. At depth 1 of the tree, there is a 90% probability that taking a left child or right child action will simply lead to the agent staying in place. Episodes are of a maximum length of 10 with $\gamma = 0.9$ for both the discount and future geometric sampling parameter (Figure 25).

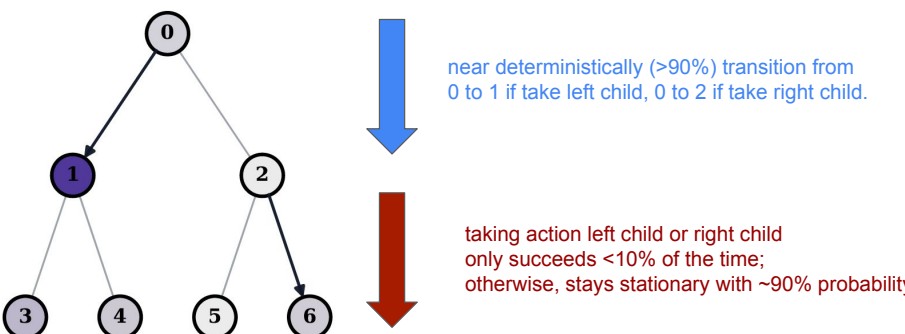

Figure 25: **Toy Tree MDP.** At the root, transitions are near deterministic. At level 1, taking the left child or right child action works only 10% of the time, while action stay works 90% of the time.

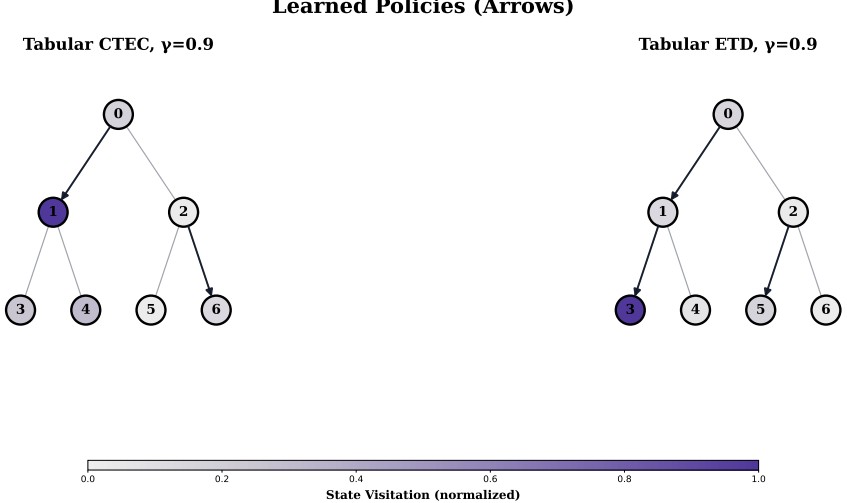

Figure 26: **forward-looking (C-TeC) vs backward-looking (ETD) rewards** As indicated by the large state visitation C-TeC prefers to stay at node 1 since it can reach multiple distinct future states, while ETD prefers to stay at node 3, the deepest node in the tree.

By definition, temporally distant states take a long time to reach. A backward-looking agent will be rewarded for visiting these states even if it has already visited them before, which can be a problem, as the agent might spend valuable training time repeatedly visiting those regions. In the aforementioned toy MDP, we expect that maximizing the backward-looking reward will incentivize the agent to continually push down the tree, and this is indeed the result in practice. On the other hand, we expect that maximizing the forward-looking reward will incentivize the agent to stay, for example, at state 1 or at the root, as the agent is rewarded for being in a state that can reach distinct future states.

To test this hypothesis, we ran C-TeC and ETD on the aforementioned MDP. We show the results in Figure 26, where arrows display the optimal learned policy after 20k episodes and color intensity denotes normalized state visitation in the last 1k episodes of training. Our results reflect the behavior of forward-looking (C-TeC) and backward-looking (ETD) policies: C-TeC prefers to stay at decision-making node 1 since it can reach diverse future states from it, while ETD tries to stay at the deepest node.

## L    COMPARISON TO MODEL-BASED BASELINES

We compare C-TeC to a model-based based on Stadie et al. (2015) and we show the results in Table 8, C-TeC outperforms model-based exploration and explore more states.

| Environment | C-TeC | MBRL (Stadie et al., 2015) |
|---|---|---|
| Ant-hardest-maze | $2500 \pm 300$ | $849 \pm 63$ |
| Humanoid-u-maze | $230 \pm 40$ | $31 \pm 8$ |
| Arm-binpick-hard | $135000 \pm 10000$ | $35000 \pm 3170$ |

Table 8: Sample Efficiency of C-TeC

## M    EMERGENT EXPLORATION BEHAVIOR

Figure 27 shows some of the learned behaviors of C-TeC in the `humanoid-u-maze`, where the agent learns to jump over the wall to escape the maze.

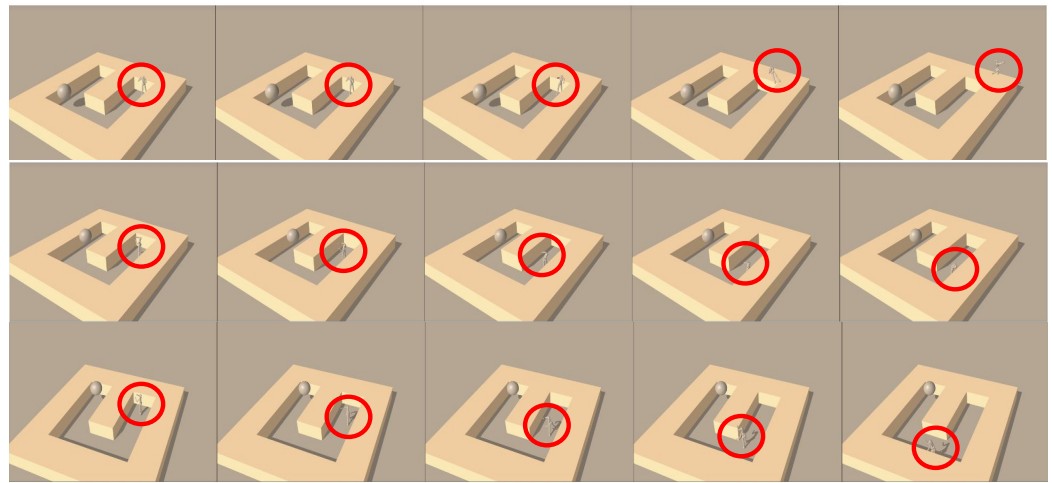

Figure 27: **Emergent Exploration Behavior in `humanoid-u-maze`.** C-TeC exhibits interesting emergent behaviors; for example, in the `humanoid-u-maze` environment, the agent learns to jump over the maze walls to escape the maze. Each row represents an independent evaluation epsidoe.

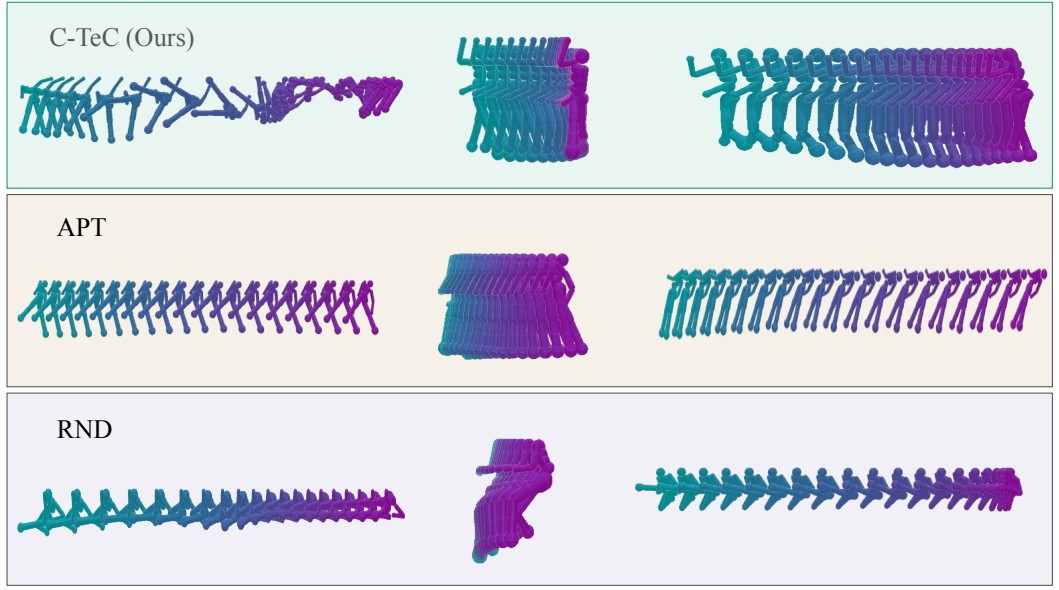

Figure 28: **Qualitative Comparison in `humanoid-u-maze`.**

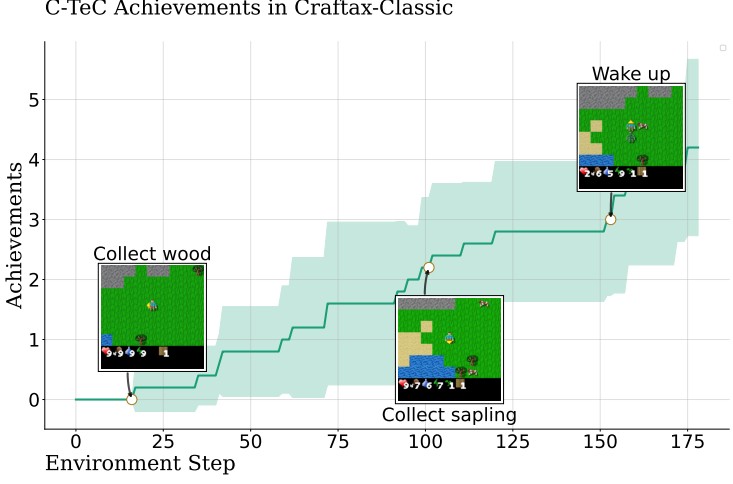

Figure 29: **C-TeC Achievements.** C-TeC unlocks interesting achievements in Craftax-Classic; the plot shows a subset of the unlocked achievements during an evaluation episode.

