# OpenReview forum: "Temporal Representations for Exploration: Learning Complex Exploratory Behavior without Extrinsic Rewards"
_ICLR.cc/2026/Conference — ICLR 2026 Poster_

### Official Review · Reviewer_yEZ5 · 2025-10-29

**Soundness:** 2
**Presentation:** 3
**Contribution:** 2
**Rating:** 6
**Confidence:** 4

**Summary:**

Based on the work of Jiang et al. (2025), this paper proposes a new reward objective that avoids quasi-metric learning and episodic memory. The experiments are conducted in navigation, manipulation, and open-world environments.

**Strengths:**

1. This paper introduces a reward objective that is easier to compute.
2. The experiments demonstrate that the proposed method leads to the agent visiting more states.

**Weaknesses:**

1. Temporal Similarity as Intrinsic Reward:

The use of temporal similarity as an intrinsic reward is explored in [1]. The novelty of this approach needs further clarification.

2. Advantages Over [1]:

Compared to [1], the proposed method (1) avoids quasi-metric learning and (2) eliminates the need for episodic memory. What are the specific benefits of these two design choices? A more detailed explanation would be helpful.

3.  Experimental Results and Task Rewards:

The experimental results show that the visited states are more diverse. However, how does this diversity benefit RL training with task rewards?
I believe this is not a significant burden on the authors, as they could simply incorporate the extrinsic reward into the current intrinsic reward framework. The absence of this experiment makes it difficult for readers to fully believe in the method's adaptability.

4. Title Clarification:

The paper is titled "DIVERSE BEHAVIORS," which typically implies that the agent can generate diverse trajectories, not just more diverse visited states. This could be somewhat confusing.


[1] Yuhua Jiang, Qihan Liu, Yiqin Yang, Xiaoteng Ma, Dianyu Zhong, Hao Hu, Jun Yang, Bin Liang, Bo XU, Chongjie Zhang, and Qianchuan Zhao. Episodic novelty through temporal distance. In The Thirteenth International Conference on Learning Representations, 2025.

**Questions:**

See above

---

> ### Author Response · Authors · 2025-11-27
> **Response to yEZ5**
>
> Dear Reviewer,
>
> Thank you for the feedback and for providing valuable suggestions to improve the paper. We revised the text and clarified the differences between our method and [1], and we updated the paper's title.
>
> > Novelty and Advantages over [1]
>
> The novelty of our method over [1] is the use of temporal similarity over future states, while [1] computes intrinsic rewards over past states. This difference results in a performance benefit, especially in Crafter (Figure 3 in the paper), and removes the need for episodic memory. We believe this design choice is important, particularly for long-horizon environments. In Appendix K, we present a toy example that clarifies when our reward is preferable to the reward in [1], especially in environments with transitions that involve arbitrarily long waiting times.
>
> Moreover, our method can work with both on-policy (PPO) and off-policy RL (SAC) algorithms, which is not demonstrated in [1].
> In addition, our method is simpler and does not require MRN architecture or learning a quasimetric, which makes it easier to implement.
>
> > Title Clarification
>
> We agree and would like to improve the clarity with a better title. We will discuss with the AC to update the title to "Temporal Representations for Exploration: Learning Complex Exploratory Behavior without Extrinsic Rewards." Our algorithm learns complex exploratory behavior without relying on extrinsic task information, unsupervised skill learning, or world-modeling. We believe that a new title clarifies the paper's main claim: that temporal contrastive representations are useful for deriving exploratory behavior.
>
> We are still working on an experiment to show how to integrate the method with a task reward, and we will post our results as soon as we have them.
>
> Do these changes address the reviewers' feedback? If not, we look forward to continuing the discussion!
>
> Best,
>
> The authors

---

### Official Review · Reviewer_aRUG · 2025-10-30

**Soundness:** 2
**Presentation:** 3
**Contribution:** 2
**Rating:** 6
**Confidence:** 3

**Summary:**

C-TeC uses a temporal contrastive critic to give intrinsic reward as the negative similarity between (s_t,a_t) and discounted future states—favoring states with diverse, hard-to-predict futures, without episodic memory and friendly to off-policy RL. It plugs into PPO/SAC, can target specific future-state subspaces, improves coverage in Ant/Humanoid mazes, and beats ETD on Crafter. Contributions: a simple contrastive exploration objective, its info-theoretic view, and consistent empirical gains across domains.

**Strengths:**

- Provides an information-theoretic interpretation: in the idealized limit, the intrinsic reward equals $−D_{KL}​(p_T​(s_f​∣s_t​,a_t​)∥p_T​(s_f​))$, linking the objective to mode-seeking over future-state distributions.

- Clear algorithmic specification (sampling of future offsets, InfoNCE update, PPO/SAC integration) that is implementable and off-policy friendly.

- Demonstrates flexible targeting of exploration to task-relevant subspaces, a practical advantage for complex embodiments and object-centric tasks

**Weaknesses:**

- The “future-diversity” view overlaps conceptually with recent contrastive or mutual-information–based exploration (e.g., APT/CIC/Plan2Explore/APS). The paper doesn’t clearly isolate what is new beyond replacing episodic memory with a temporal contrastive critic

- The intrinsic reward hinges on design knobs that may materially affect behavior: future horizon/window sampling, temperature in InfoNCE, negative-sampling strategy, and choice of future subspace (e.g., positions vs. full features)

- Using negative similarity as reward risks mode chasing or collapse if the representation is not sufficiently regularized.

**Questions:**

- You connect the objective to $−D_{KL}​(p_T​(s_f​∣s_t​,a_t​)∥p_T​(s_f​))$ or $I((s_t,a_t);s_f)$, under what assumptions (temperature τ, negative sampling, batch size) does the empirical contrastive loss yield a consistent estimator of this quantity? Can you give a finite-sample bound relating representation error 𝜖 to the bias in $r_{int}$?

- How do you choose the distribution over offsets k∼p(k) for the “discounted future”? Did you try adaptive or curriculum schedules for k (e.g., increase max k over training)? Please add sensitivity/ablation for p(k) and the discount used inside the similarity.

- Contrastive objectives can be biased by replay imbalance (e.g., many near-duplicate futures from recent policies). Do you stratify negatives by time/episode, or reweight by importance? Ablate (i) in-episode vs. cross-episode negatives, (ii) balanced vs. raw replay sampling, and report stability/sample-efficiency impacts.

---

> ### Author Response · Authors · 2025-11-21
> **Response to aRUG**
>
> Dear Reviewer,
>
> Thank you for the detailed feedback and for providing valuable suggestions to improve the paper. We have incorporated the feedback through highlighting the similarities and differences between our work and prior works. We ran ablations on the design knobs of our method. We also investigated the effect of in-episode vs across-episode negative sampling.
>
> > The “future-diversity” view overlaps conceptually with recent contrastive exploration (e.g., APT/CIC/Plan2Explore/APS)
>
> We have revised the related work section to highlight these similarities while also noting that these prior works are designed for different problem settings or use contrastive learning in different ways. APS and CIC are skill learning algorithms, while our method is a pure exploration method. APT is also a pure exploration method that uses contrastive learning to learn representations from image-based observations, but it does not encode the temporal structure of the environment, which results in its lower performance w.r.t C-TEC. Plan2Explore uses the prediction error from a world model as the exploration signal, whereas our method does not require learning a world model.
>
> > The intrinsic reward hinges on design knobs that may materially affect behavior: future horizon/window sampling, temperature in InfoNCE, negative-sampling strategy, and choice of future subspace
>
> We ran several ablation studies to understand the effect of various design knobs, and due to space limitations, we placed them in Appendix G. We found that representation normalization, the temperature in InfoNCE, and the contrastive similarity function are three important design choices to which our method is particularly sensitive.  Choosing a subset of the future state makes the exploration space smaller, which results in better performance. However, our method still outperforms the baselines in both settings (Figure 10).
>
> > How do you choose the distribution over offsets k∼p(k) for the “discounted future”?
>
> We found that a discount value of 0.99 worked well in most environments (expected offset ≈ 100). To assess sensitivity, we ablated the future-state sampling strategy, including a linear discount schedule; the results (Figure 16, Appendix G.6) show the method is robust to this choice.
> We also ablated different fixed discount values (in geometric sampling of the future state). As shown in Figure 17 (Appendix G.7), 0.99 provides strong average performance, though tuning the discount per environment can further improve results.
>
> > Using negative similarity as a reward risks mode chasing or collapse
>
> We plot the variance of the contrastive representations (see appendix C, Figure 9) throughout training. The plot shows that the variance of the representations is consistently greater than zero, which indicates that our method does not suffer from mode collapse.
>
> >  You connect the objective to $\ D_{KL}(p_T(s_f \mid s_t,a_t) || p_T(s_f))$   or $I((s_t,a_t);s_f)$ under what assumptions (temperature τ, negative sampling, batch size) does the empirical contrastive loss yield a consistent estimator of this quantity? Does the empirical contrastive loss yield a consistent estimator of this quantity? Can you give a finite-sample bound relating representation error 𝜖 to the bias in kl?
>
> In our experiment, the temperature $\tau$ is treated as a learned parameter, In the negative sampling, for each batch, the positive examples of other samples are treated as negatives, following common practice [1]. We use the same batch size of 1024 across the robotic environments.
> We refer the reviewer to [2] for a detailed discussion on bounds, consistency and statistical efficiency of contrastive models.
>
> > Ablate (i) in-episode vs. cross-episode negatives, (ii) balanced vs. raw replay sampling, and report stability/sample-efficiency impacts.
>
> We ablated the negative sampling strategy (in-episode vs. cross-episode). Following [3], we control the number of in-episode negatives by duplicating trajectories in the batch using a repetition factor (1 = cross-episode; larger values = more in-episode negatives). Figure 19 shows that in-episode sampling improves exploration performance and sample efficiency, while stability remains unchanged.
> Regarding balanced vs. raw replay sampling, could the reviewer clarify which notion of “balance” they are referring to?
>
> [1] Chen, Ting, et al. "A simple framework for contrastive learning of visual representations." ICML PmLR, 2020
>
> [2] Ma, Zhuang, and Michael Collins. "Noise Contrastive Estimation and Negative Sampling for Conditional Models: Consistency and Statistical Efficiency." Proceedings of the 2018 Conference on Empirical Methods in Natural Language Processing. 2018.
>
> [3] Ziarko, et al. "Contrastive Representations for Temporal Reasoning." The Thirty-ninth Annual Conference on Neural Information Processing Systems.
>
> Do these changes address the reviewers' feedback? if not, we look forward to continuing the discussion!
>
> Best,
>
> The authors

---

> > ### Author Response · Authors · 2025-11-27
> >
> > Dear Reviewer,
> >
> > We hope that you've had a chance to read our responses and clarification. As the discussion period is approaching its end, we would greatly appreciate it if you could confirm that our updates have addressed your concerns.
> >
> >
> > Best,
> >
> > the authors

---

### Official Review · Reviewer_xkVH · 2025-11-01

**Soundness:** 3
**Presentation:** 3
**Contribution:** 3
**Rating:** 6
**Confidence:** 5

**Summary:**

This paper focuses on the problem of using intrinsic motivation to promote exploration in reinforcement learning, primarily by designing a new intrinsic reward. This new intrinsic reward directly follows the previous work "Episodic Novelty Through Temporal Distance (ETD)." The main motivation is to simplify the ETD method, with the following improvements:

- ETD learns the temporal distance between states.  Distance meaning it should at least satisfy the properties of a quasi-metric. C-TEC demonstrates through experimental results that learning a quasi-metric is unnecessary; it is sufficient as long as the representation can reflect the temporal relationship between states.
- ETD calculates the intrinsic reward by computing the distance between the current state and every state in the episodic memory, taking the minimum value as the intrinsic reward (backward-looking). In contrast, C-TEC samples a future state according to a geometric distribution and calculates the distance between the current state and that future state as the intrinsic reward (forward-looking).

**Strengths:**

The algorithm uses the idea of temporal distance as an intrinsic reward. It has some contributions in simplifying ETD's algorithmic modules, and it was found on continuous control tasks and the Crafter task that simplifying these modules does not lead to performance degradation. I believe this contribution is sufficient.

Changing ETD's intrinsic reward from backward-looking to forward-looking is interesting.

**Weaknesses:**

- The comparison results between ETD and C-TEC on Crafter are very strange. In the original ETD paper, the results for the method and its baselines were all in the (7.0 ~ 9.0) range. However, here the result for ETD is less than 0.5, and even C-TEC is only 2.0.

- The paper's title, "Discovering Diverse Behaviors via Temporal Contrastive Learning," is somewhat strange. I believe "Discovering Diverse Behaviors" corresponds to learning a set of policies, each with a different behavior. However, the paper is clearly not doing this. This title is ambiguous.

- In line 256, the paper divides the intrinsic reward into a "surprise" term and a "familiarity" term. The "surprise" term can be understood as promoting exploration, but I do not understand the role of the "familiarity" term. I also do not quite understand the role of "mode-seeking" in exploration. Does the author mean that a high reward should be given to states that have been visited but are temporally distant, rather than giving a high reward to unvisited states?

- C-TEC is (forward-looking) while ETD is (backward-looking). Besides the fact that the forward-looking approach does not require an episodic memory, are there any other advantages compared to the backward-looking approach? I find the current Crafter results unconvincing, and I believe that results from some toy examples would be more illustrative.

**Questions:**

See Weakness

---

> ### Author Response · Authors · 2025-11-20
> **Response to xkVH**
>
> Dear Reviewer,
>
> Thank you for the feedback on improving the paper. We clarified the discrepancy in the ETD results and revised the corresponding text, updated the paper’s title, clarified the mode-seeking view of our exploration objective, and further clarified the differences between ETD and C-TEC.
>
> > The comparison results between ETD and C-TEC on Crafter are very strange.
>
> To clarify, our experiments use only intrinsic rewards, whereas the results in the ETD paper use a combination of intrinsic and extrinsic rewards. This design difference explains the discrepancy between the results reported in our paper and those in the ETD paper (which also used an extrinsic reward). Our primary claim is that C-TEC can learn interesting and useful behaviors through our temporal contrastive objective, even in the complete absence of task rewards.
>
> > The paper's title, "Discovering Diverse Behaviors via Temporal Contrastive Learning," is somewhat strange.
>
> We agree and would like to improve the clarity with a better title. We will discuss with the AC to update the title to "Temporal Representations for Exploration: Learning Complex Exploratory Behavior without Extrinsic Rewards." Our algorithm learns complex exploratory behavior without relying on extrinsic task information, unsupervised skill learning, or world-modeling. We believe that a new title clarifies the paper's main claim: that temporal contrastive representations are useful for deriving exploratory behavior.
>
> >  I also do not quite understand the role of "mode-seeking" in exploration. Does the author mean that a high reward should be given to states that have been visited but are temporally distant, rather than giving a high reward to unvisited states?
>
> Yes, we mean that a high reward should be given to states that have been visited but are temporally distant, rather than giving a high reward to unvisited states. We have revised the section (5.1) in the paper to make this idea clearer. We refer to section 6.2.6 in [2] for a more detailed explanation of the reverse KL and mode-seeking behavior.
>
> > Comparing C-TEC to ETC, are there any other advantages compared to the backward-looking approach?
>
> Regarding the advantages of C-TEC over ETD, in addition to the distinction between forward- and backward-looking rewards, ETD requires a more constrained architecture—specifically the MRN [1] to learn quasimetric representations that encode temporal distance. Our findings show that such architectural constraints are not necessary for effective exploration. Learning representations that capture the temporal structure of policy behavior and environment dynamics is sufficient to achieve meaningful exploration without the MRN architecture or temporal distances. Moreover, as the reviewer mentioned, our method does not require episodic memory, and it does not need to take the minimum over the distance among past states, which makes our method easier to implement while having similar or better performance than ETD. We have incorporated the feedback through editing the paper’s section about ETD and C-TEC results. And clarifying the advantages of C-TEC over ETD.
>
> [1] Liu, Bo, et al. "Metric residual network for sample efficient goal-conditioned reinforcement learning." Proceedings of the AAAI Conference on Artificial Intelligence. Vol. 37. No. 7. 2023.
>
> [2] Murphy, Kevin P. Probabilistic machine learning: an introduction. MIT press, 2022.
>
> **Do these changes, together with the additional revisions and clarifications discussed below, fully address the reviewer's feedback about the paper?** If not, we look forward to continuing the discussion!
>
> Kind regards,
>
> The authors

---

> ### Comment · Reviewer_xkVH · 2025-11-21
>
> Thank the authors for their response.
>
> If the authors intend “mode-seeking” to mean that “a high reward should be given to states that have been visited but are temporally distant, rather than to unvisited states,” please provide a targeted experiment to substantiate the benifit of this claim; the Crafter results are not particularly convincing.
>
> At present, I do not see any advantage of the backward-looking approach over a forward-looking one beyond removing the need for episodic memory. The purported “mode-seeking” effect appears more like a post-hoc theoretical justification than an empirically demonstrated benefit.

---

> > ### Author Response · Authors · 2025-11-25
> > **Response to xkVH**
> >
> > Dear Reviewer,
> >
> > Thank you for your questions.
> >
> > In response to the reviewer requests for didactic experiments + more clarity, we ran two toy examples that we believe strengthens the claims of the paper: one that addresses the **forward vs. backward comparison** (Experiment A) and another that addresses the **familiarity/mode-seeking term in the intrinsic reward** (Experiment B). The experiments show that CTEC and ETD can behave very differently and optimize different objects. All conclusions are consistent over 3 seeds.
> >
> > Throughout, we drop the representation learning given the simple tabular settings. Instead, we directly fit the intrinsic MI objects by maintaining tabular counts of state occupancies; this is for simplicity and clarity of the following takeaways.
> >
> > We hope that the experiments and accompanying explanation clarifies the differences between CTEC and ETD. We will add these results to the paper. **Do the following experiments and explanations address the reviewers concerns and questions?**
> >
> > ______
> > # Experiment A: On forward vs. backward directions
> >
> > ## Forward-looking reward avoids unnecessarily repeating time-consuming behaviors
> > > C-TEC is (forward-looking) while ETD is (backward-looking). Besides the fact that the forward-looking approach does not require an episodic memory, are there any other advantages compared to the backward-looking approach? [...] I believe that results from some toy examples would be more illustrative [...]
> >
> > Forward-looking policies are beneficial when there are **transitions with arbitrarily long waiting times**. We present some conceptual reasoning and simple results in toy examples as requested by the reviewer.
> >
> > By definition, temporally distant states take a long time to reach. A backward-looking object rewards the agent for reaching these states **even if there’s already many previous examples of the same trajectories**. This can be a problem because previously-seen transitions can have arbitrarily long waiting times.
> >
> > For example, suppose an agent can transition from state A -> state B if it (1) finds a 100-sided die and (2) rolls a value of 1. A policy that rewards backward temporal distance will continue to find rolling the die rewarding, even after it's seen the 1 many times. While state B is very temporally distant, during exploration, we should sometimes care more about the agent **acquiring the die** than incentivizing the agent to continually roll for a 1, which wastes training time.
> >
> > We show this phenomena in a simple experiment.
> >
> > ## Result of Experiment A: forward and backward are different ##
> >
> > Consider a binary tree MDP with 3 layers. Let action space $\mathcal{A} = \\{\text{left child}, \text{right child}, \text{stay}\\}$. Actions are fully deterministic from the root, with the exception of a 10% chance of a random action at any node. Episodes are of max length 10 and begin at the root.
> >
> > At depth 1 of the tree (depth 0 indexes the root), there is a **~90% probability** that taking a left child or right child action will lead to the agent **staying in place**; there is also a 10% chance at all times to effectively take a random action for extra stochasticity, which slightly perturbs the probabilities.
> >
> > The transition probabilities make the leaf nodes temporally distant from the nodes at depth 1 and the root node. The node at depth 1 and the root node are temporally close. There is no way to go up the tree, only down. See https://anonymous.4open.science/r/rebuttal_plots-17B6/MDP%20explanations.pdf for a visual summary of the MDP.
> >
> > We run ETD and CTEC and find the following:
> >
> > (1) **Maximizing backward temporal distance incentivizes agent to continually push down the tree.** See results here https://anonymous.4open.science/r/rebuttal_plots-17B6/policies_comparison_tree_seed=0.pdf
> > (right, ETD policy pushes down the tree for most of the time), where arrows display the optimal learned policy after 20k episodes and color intensity denotes normalized state visitation in the last 1k episodes of training.
> >
> > (2) **Maximizing forward temporal distance (like in CTEC) incentivizes agent to stay at depth 1 of the tree**. The agent is rewarded for being at states that **can** reach distant states (leaves), but do not necessarily need to reach said states within a trajectory to gain reward. The resulting learned policy and state visitation frequencies reflect this behavior:
> > https://anonymous.4open.science/r/rebuttal_plots-17B6/policies_comparison_tree_seed=0.pdf
> > (left, CTEC policy stays at depth 1 of the tree).
> >
> > Thus, after figuring out the long temporal distance between the root node and leaf nodes, the forward-looking policy does not bother to keep repeating these time-consuming behaviors. An effective exploration agent could, instead, reach many states where such long-horizon trajectories are known to be possible.
> >
> > In conclusion, forward vs. backward temporal distance objectives achieve different but important aims and are clearly separated in this simple MDP.

---

> ### Author Response · Authors · 2025-11-25
> **Continuation of response to xkVH**
>
> # Experiment B: On the interpretation of the intrinsic reward
>
> > In line 256, the paper divides the intrinsic reward into a "surprise" term and a "familiarity" term. The "surprise" term can be understood as promoting exploration, but I do not understand the role of the "familiarity" term. I also do not quite understand the role of "mode-seeking" in exploration. Does the author mean that a high reward should be given to states that have been visited but are temporally distant, rather than giving a high reward to unvisited states?
>
> We believe the following didactic toy example (Experiment B) and explanation clarifies the interpretation of the intrinsic reward. We begin with mathematical motivation for the experiment, then explain the MDP. The linked figures make the MDP setup clearer. All conclusions are consistent over 3 seeds.
>
> ## Difference between MI objects in CTEC and ETD
>
> “Mode-seeking” means exactly what the reviewer wrote: high reward is given to states that are distant but still have representation in the buffer, rather than unvisited or rarely-visited states. This is different from temporal distance and leads to qualitatively different effects.
>
> Focusing solely on the MI objects, temporal distance (á la ETD) maximizes:
>
> $$ \mathbb{E}[r_{\\text{ETD}}] = \mathbb{E}[\\log (p(s_+ \\mid s_+)/p(s_+ \\mid s,a))]. $$
>
> Thus, ETD rewards states that look improbable ($p(s_+ \mid s,a)$ is small) relative to local reachability $ p(s_+ \mid s_+).$ If $p(s_+)$ has really small representation within the buffer but is easily reachable from $s_+$ (e.g. $s_+$ is an absorbing state), the expected temporal distance becomes large. Term $p(s_+ \mid s_+)$ captures a form of **local reachability** – reward states that are reachable locally/from itself, but tough to reach from my current state-action.
>
> Meanwhile, the CTEC objective maximizes:
>
> $$ \mathbb{E}[r_{\text{CTEC}] = \mathbb{E}[\log (p(s_+)/p(s_+ \mid s,a))] $$
>
> The familiarity term comes from the numerator $\log p(s_+)$: we’re rewarding states that look improbable (small $p(s_+ \mid s,a)$) relative to the marginal $p(s_+)$. This term captures **global reachability** – reward states that are reachable globally, but tough to reach from my current state-action.
>
> ## Result of Experiment B: CTEC and ETD treat reachability differently
>
> The following experiment shows this local vs. global reachability effect clearly. See (https://anonymous.4open.science/r/rebuttal_plots-17B6/MDP%20explanations.pdf) for a figure of the MDP and details on the action space.
>
> Consider an MDP where an agent starts from a fixed node (14) and must immediately decide whether to commit to a left subgraph (0-6) or a right subgraph (7-13). Both sides consist of individually fully connected graphs between states. There is no way to transition from one side to another. Actions generally consist of specified nodes to transition to from other nodes in the subgraph (see fig. for more details). Episodes are of length 70.
>
> Importantly, the **right branch has ``sticky’’ transitions**: with ~90% probability, the agent stays at its current node regardless of the action. For a given action, it transitions to the specified node ~10% of the time; there is also a 10% chance at any point to take a random action for extra stochasticity, which slightly perturbs the probabilities. All states are locally reachable (local reachability term $\log p(s_+ \mid s_+)$ in ETD is high, because the states are ``sticky''), but are not globally reachable (familiarity term $\log p(s_+)$ in CTEC is generally low).
>
> Meanwhile, the **left branch has freely-moving transitions**. The action space consists of $\mathcal{A} = \\{\text{go to node i}, \text{go to random node}\\}$, where nodes **cannot** transition to themselves. However, the branch is very dynamic -- valid actions $a = \text{go to node i}$ will successfully transition with ~100% probability. **Because self-transitions are not allowed,** all states are globally reachable (“familiarity” term from $\log p(s_+)$ in CTEC can easily be high), but are not locally reachable (term $\log p(s_+ \mid s_+)$ in ETD is low – cannot stay in the same state over timesteps).
>
> Thus, we should expect that the mode-seeking **CTEC reward will choose the left branch, prioritizing global reachability** (what we call ``mode-seeking'' in the paper), and that the temporal distance **ETD reward will choose the right branch, prioritizing local reachability**. Experiments show this clear left/right split between CTEC and ETD, showing that the fundamental MI objects lead to different, mathematically-supported exploration behaviors:
>
> CTEC (left, https://anonymous.4open.science/r/rebuttal_plots-17B6/policies_comparison_split_seed=0.pdf)
>
> ETD (right, https://anonymous.4open.science/r/rebuttal_plots-17B6/policies_comparison_split_seed=0.pdf)

---

> > ### Author Response · Authors · 2025-11-25
> > **Summarizing above**
> >
> > In summary, we set up and ran two experiments clarifying differences between CTEC and ETD:
> > (1) Experiment A:
> > - forwards-in-time (CTEC) vs. backwards-in-time (ETD) optimize different objects that leads to markedly different and interpretable behavior in a simple tree MDP.
> > - we may prefer forwards-in-time rewards to avoid repeating time-consuming, known trajectories, and, instead, prioritize reaching states that **can** get to these long trajectories (e.g. acquiring and repeatedly rolling a many-sided die vs. just acquiring a many-sided die).
> >
> > (2) Experiment B:
> > - CTEC rewards tough-to-reach states relative to global reachability $\log p(s_+)$. Meanwhile, ETD rewards tough-to-reach states relative to local reachability $\log p(s_+ \mid s_+)$. These are not the same and lead to divergent behavior in two-pronged MDP, where one branch prioritizes one effect over the other.
> > - We agree that the ``familiarity'' description of this effect is confusing, and will revise the paper accordingly now supported by these didactic experiments.

---

> ### Comment · Reviewer_xkVH · 2025-11-26
>
> Thank you for the explanation. The distinction is clear now: backward-looking implies depth optimization, whereas forward-looking favors key decision nodes.
>
> However, I am still puzzled by the underlying mechanics of ETD:
>
> First, regarding the objective $\mathbb{E}[r_{\text{ETD}}] = \mathbb{E}[\log (p(s_+ | s_+)/p(s_+ \mid s,a))]$, is the goal of **minimizing episodic memory distance** explicitly ignored here?
>
> Second, I find the definition of ETD as "local reachability" (and CTEC as "global") counterintuitive. Since ETD is designed to minimize temporal distance to valid states, logically it should encourage the agent to move away from current states rather than staying put. Therefore, it seems ETD should **not** encourage "sticky" behaviors.

---

> > ### Author Response · Authors · 2025-11-29
> > **Response to vkVH concerns and questions**
> >
> > In response to the reviewer concerns and questions, we do the following:
> >
> > (1) explain the episodic novelty component of ETD,
> >
> > (2) revise and simplify the setting of Experiment B to show CTEC and ETD are maximizing fundamentally different objects,
> >
> > (3) present a simple empirical argument for why the $r_{\\text{intr}}$ formulations of CTEC and ETD lead to different behaviors in Experiment B, and
> >
> > (4) clarify the global/local reachability statement using the results as support.
> >
> > > The distinction is clear now: backward-looking implies depth optimization, whereas forward-looking favors key decision nodes.
> >
> > We are glad that Experiment A clarified the differences between forward and backward approaches, one of the main differences between CTEC and ETD.
> >
> > > is the goal of minimizing episodic memory distance explicitly ignored here?
> >
> > No. In all of the presented experiments, both here and in the paper, we are **maximizing the discounted sum of the minimum episodic memory distance** like in Jiang et al. (2025).
> >
> > ETD **maximizes (via the Q function) the (summed) minimum distance between episodic states and the current state**. This minimization over episodic states is to ensure temporal distance is measured with respect to an ensemble of seen states. Optimizing backwards temporal distance from the previous state to current state, or any type of fixed prior state, can lead to degenerate behavior like flipping back-and-forth between two temporally-distant states.
> >
> > However, an agent maximizing ETD is **still maximizing a temporal distance**. CTEC is maximizing a different object. The maximization of these two objects can lead to a divergence between ETD and CTEC agents with or without the inner episodic minimization.
> >
> > The following results show evidence of this in a very simple two-armed bandit setting. See link for the MDP figure (https://anonymous.4open.science/r/rebuttal_plots-17B6/MDP_explanations.pdf):
> > - The MDP consists of a root node connected to a right and a left branch. All trajectories begin from this root node.
> > - The left branch contains fast dynamics: the agent deterministically moves down the branch to the leaf node, progressing one level per timestep.
> > - The right branch contains slow/sticky dynamics. With $90\%$ probability, the agent will remain stuck at the state for a given timestep. With $10\%$ probability, the agent will progress down the tree.
> > - **The dynamics of the agent are completely independent of any policy** after choosing the branch. Thus, **this problem is effectively a 2-armed bandit** where the agent needs to choose which branch maximizes the objective.
> > - We consider episodes of length 30 and $\\gamma=0.99$ for discount and future state sampling in CTEC.
> >
> > An agent maximizing the CTEC objective wants to match the marginals $p(s_+)$ and $p(s_+ \\mid s,a)$ – one can see this from the mode-seeking KL formulation or from the familiarity term $\\mathbb{E}\_{p(s_+ \\mid s,a)}[\\log p(s_+)]$.
> >
> > We visualize the future state distributions for different nodes: https://anonymous.4open.science/r/rebuttal_plots-17B6/future_state_distributions_racing_stripes.pdf in the left (left figure) and right (right figure) branches (see second page of https://anonymous.4open.science/r/rebuttal_plots-17B6/MDP_explanations.pdf for MDP structure). Clearly, the left side of the MDP leads to $p(s_+ \\mid s,a)$ that much more readily matches the marginal $p(s_+)$: the state visitation has a mode at leaf node 2 with very little probability mass on 0 and 1 in the buffer.
> >
> > Meanwhile, the right branch assigns more visitation probability to 3 and 4 from the slow dynamics: the distributions $p(s_+ \\mid s,a)$ are less aligned with the marginal.
> >
> > Correspondingly, the CTEC agent chooses the **left, fast-moving** branch, reflecting our mode-seeking interpretation, and the ETD agent chooses the **right, slow-moving/sticky** branch where nodes are temporally distant (see figure https://anonymous.4open.science/r/rebuttal_plots-17B6/policies_comparison_racing_stripes.pdf, state visitations of last 1000 episodes and learned policies). **Even though the MDP structure on the left and right are identical, the difference in dynamics splits the methods’ behaviors.**
> >
> > The interpretation of these results is below.

---

> ### Author Response · Authors · 2025-11-29
> **continued**
>
> > Second, I find the definition of ETD as "local reachability" (and CTEC as "global") counterintuitive. Since ETD is designed to minimize temporal distance to valid states, logically it should encourage the agent to move away from current states rather than staying put. Therefore, it seems ETD should not encourage "sticky" behaviors.
>
> By local and global reachability, we are referring to the MDP dynamics that the agent prefers rather than the agent dynamics themselves. You are right that ETD does not encourage ``sticky’’ behaviors **within a responsive MDP** – for example, ETD pushes down the tree in Experiment A instead of staying still at the root.
>
> However, given a choice between **identical MDP structures with fast-moving or sticky/slow dynamics**, ETD-maximizing agents **can have a clear preference for the slow-moving MDP dynamics**. This is evidenced by the ETD agent choosing the right branch of Experiment B to maximize temporal distance: nodes directly connected by a single sticky transition are close in state space/graph space but are very distant in temporal space. In the case of Experiment B, this is irrespective of whether distances are measured relative to an ensemble (episodic minimization) – here, temporal distance monotonically increases with graph distance and the agent can only traverse down the tree – and is a function of the intrinsic reward probability ratio.
>
> Correspondingly, if you looked at a video of an agent traversing down the branches in Experiment B, the CTEC agent would reach a leaf node much faster. This is not because the agent is inherently ``faster’’ than the ETD agent: instead, this is because the intrinsic reward objects inherently favor **different subgraphs of the MDP with different dynamics**.
>
> While we used the terms global and local reachability within the reviewer responses, in paper revisions, we will be more careful to specify that global/local reachability is a comment on the **MDP dynamics preferred by the CTEC/ETD-maximizing agent** (assuming otherwise identical MDP structures) rather than an inherent property of the agents themselves.
>
> Finally, we’d like to note that this experiment does not contradict the results of Experiment A: here, we explicitly do not allow the agent to stay still at key decision nodes – the agent must always push down the tree.
>
> ______
>
> In summary, we simplify Experiment B to show that agents maximizing CTEC and ETD have different preferences for MDP dynamics irrespective of the inner episodic minimization: CTEC has a preference for MDP dynamics that more readily enable state-conditioned future probabilities to seek globally-reachable modes, and ETD has a preference for MDP dynamics that maximize the achievable temporal distance.
>
> In Experiment B, this manifests as CTEC choosing the fast-moving, deterministic branch, and ETD choosing the slow-moving, sticky branch of the MDP.

---

### Official Review · Reviewer_6M1w · 2025-11-04

**Soundness:** 3
**Presentation:** 3
**Contribution:** 3
**Rating:** 8
**Confidence:** 2

**Summary:**

This article proposes C-TeC (Curiosity via Temporal Contrastive Learning), a novel exploration method that relies on intrinsic rewards and contrastive learning to reach unexpected but meaningful states. This technique differentiates itself from prior methods by its simplicity, avoiding the use of quasimetric learning. The method is also compatible with classic algorithms such as Soft Actor-Critic and Proximal Policy Optimization. Finally, empirical results show strong exploration performance on locomotion, manipulation, and open-world tasks.

**Strengths:**

The presented empirical results show generally strong performance across various tasks, providing solid evidence of state-of-the-art performance.

The article presents sufficient theoretical background to justify the proposed method, going beyond simple empirical results.

The proposed approach is conceptually simple yet broadly applicable, removing complex components such as episodic memory or explicit distance metrics while maintaining competitive results. This simplicity makes C-TeC easier to integrate with standard off-policy algorithms like SAC and potentially more stable and efficient in training compared to previous exploration methods.

**Weaknesses:**

While the authors present extensive ablation studies, the paper does not include ablations related to sample efficiency or the method's sensitivity to hyperparameters.

Additional benchmark comparisons with recent state-of-the-art world-model-based algorithms, such as DreamerV3, would provide valuable insight into the trade-offs between their higher computational cost and their performance compared to the proposed method.

While the authors recognize that this will be tackled in future work, understanding the performance of the proposed method in partially observable environments is crucial to demonstrate the method’s broader applicability.

**Questions:**

-

---

> ### Author Response · Authors · 2025-11-26
> **Response to 6M1w**
>
> Dear Reviewer,
>
> Thank you for your feedback
>
> > Ablations related to sample efficiency
>
> To address this, we ran our method with 50M timesteps, which is 10X less than the reported results. Our results show that our method  can effectively explore to cover the space of useful states with fewer environment interactions, as shown in the table below:
> | Environment | 500M steps | 50M steps |
> | :--- | :--- | :--- |
> | Ant-hardest-maze | $2500 \pm 300$ | $1916 \pm 430$ |
> | Humanoid-u-maze | $230 \pm 40$ | $143 \pm 34$ |
> | Arm-binpick-hard | $135000 \pm 10000$ | $40000 \pm 14000$ |
>
> We also visualize the state coverage in the ant-hardest-maze with 50M vs 500M environment steps, and we added it to the paper in Appendix B, Figure 8.
>
>
> > Sensitivity to hyperparameters,
>
> We ran ablations on the core components of our method (representation normalization, contrastive losses, future-state sampling strategy, and the contrastive critic architecture) due to space constraints; we moved these ablations to appendix G If there are additional ablations of interest we missed, please specify the details.
>
> > The model-based baseline
>
> We ran a model-based RL (MBRL) exploration baseline based on [1] and we show the results compared to our method in the table below
> | Environment | MBRL | CTEC |
> | :--- | :--- | :--- |
> | Ant-hardest-maze | $849 \pm 63$ | $2500 \pm 300$ |
> | Humanoid-u-maze | $31 \pm 8$ | $230 \pm 40$ |
> | Arm-binpick-hard | $35000 \pm 3170$ |$135000 \pm 10000$ |
>
>
> These results show that our method is better than MBRL in covering states.
>
> Regarding a world-model baseline, most world-model-based exploration algorithms assume image-based observations, in contrast to our method, which assumes vector-based observations/states in most of our experiments, except in the comparison to ETD (Figure 3). For reference, we report the results of Plan2Explore [2] (a world-model-based exploration algorithm based on Dreamer v2) from the original Crafter paper [3]. Note that these results are based on optimizing only the intrinsic reward.
>
> | Environment | Plan2Explore | CTEC | ETD |
> | :--- | :--- | :--- | :--- |
> | Crafter| $2.1 \pm 0.1$ | $2.1 \pm 0.3$ | $0.3 \pm 0.04$
>
> These results show that our method, despite being model-free, is competitive with Plan2Explore, which is based on DreamerV2.
>
> > Understanding the performance of the proposed method in partially observable environments.
>
> We highlight our results in Craftax and Crafter, which are both partially observed environments. In Craftax, the observation is a symbolic representation of the agent’s local view, while in Crafter, the observation is an image of size 64x64x3. Our results in these two environments show that our method can work effectively (In Craftax, our method achieved an achievement rate of about 6 compared to the strongest baseline, which had an achievement rate of 4 (Figure 7 and 3 in the paper)  in a partially observed setting.
>
> Please let us know if there is anything else we can do to strengthen the paper!
>
> [1] Stadie, Bradly C. et al. “Incentivizing Exploration In Reinforcement Learning With Deep Predictive Models.” ArXiv abs/1507.00814 (2015): n. Pag.
>
> [2] Sekar, Ramanan et al. “Planning to Explore via Self-Supervised World Models.” International Conference on Machine Learning (2020).
>
> [3] Hafner, Danijar. “Benchmarking the Spectrum of Agent Capabilities.” ArXiv abs/2109.06780 (2021): n. pag.
>
> Kind regards,
>
> -- The authors

---

### Author Response · Authors · 2025-12-03
**Rebuttal summary for the area chair**

Dear AC,

We provide a summary of our work and the updates we made to address each reviewer concern.

The contribution of our work is an exploration method based on temporal contrastive learning, our new exploration objective is based on the prediction error of temporal representations that encode the agent’s discounted future state occupancy, this simple objective results in learning complex exploratory behavior in mazes to open environments like Craftax-classic (our method discovered 50% more achievements in Craftax-classic than the strongest baseline).

We thank the reviewers for their valuable feedback, helpful questions, and insightful discussions. Below, we summarize the main concerns of each reviewer along with their scores:


**Reviewer 6M1w (score: 8, confidence: 2):** The main concerns of reviewer 6M1w were: (1) the sample efficiency of our method, (2) sensitivity to hyperparameters, (3) comparison to model-based or world-model baselines, and (4) performance under partial observability.

**Reviewer xkVH01 (score: 6, confidence: 5):** The main concerns of reviewer xkVH01 were: (1) comparison to ETD , a similar exploration baseline based on maximizing temporal distance, (2) the clarity of the paper’s title, (3) interpretation of our proposed exploration reward, and (4) further clarification of forward-looking vs. backward-looking exploration rewards and the practical differences between them.

**Reviewer aRUG30 (score: 6, confidence: 3):** The main concerns of reviewer aRUG30 were: (1) conceptual overlap with prior work and how our method differs from previous contrastive learning–based exploration methods, (2) the design knobs of our intrinsic reward (future-horizon/window sampling, InfoNCE temperature, negative-sampling strategy, etc.) and how they may affect behavior, (3) the risk of mode collapse in contrastive representations, and (4) the effect of the negative-sampling strategy.

**Reviewer yEZ5 (score: 6, confidence: 4):** The main concerns of reviewer yEZ5 were: (1) the overlap between our method and [1] and the advantages of our approach, (2) how to integrate our intrinsic reward with a task reward, and (3) the clarity of the paper’s title.

Based on the reviewers' feedback, we improved our paper as follows:

**Theoretical interpretation of our exploration reward and novelty over prior work (reviewers: xkVH and yEZ5):**  We discussed the interpretation of our exploration objective with reviewer xkVH and we introduced simple toy example that gives intuitions about our proposed objective, we also added a toy example that highlight the difference between our exploration reward (forward-looking exploration) and the reward from ETD (backward looking exploration) which is a closely related work. These two toy examples highlight the novelty and advantage of our method and distinguish it from prior work. These examples are in Appendices J.2 and K. We also discussed in the related work section how our method is different from prior work that uses contrastive learning for exploration.


**Ablation study and negative sampling strategy (reviewer: aRUG):** We added an ablation study in the appendix. This study provides insight into the core components of our algorithm and how to adjust it to different environments. Specifically, we ablated components such as the type of contrastive loss, representation normalization, and the discount factor of the discounted future state occupancy. We also ablated the negative sampling strategy. We found that the method is particularly sensitive to representation normalization and the discount factor, and we found that sampling negative in an in-episode might boost performance. All of our ablations are in Appendix G.

**Data efficiency(reviewer: 6M1w ):** We ran our method with 50M timesteps, which is 10X less than the reported results in the main paper. Our results show that our method can effectively explore and cover the space of useful states with fewer environment interactions, and it also shows the scalability of our method; with more training, it explores more. the results are in Appendix B.

**Title clarification (reviewers: xkVH and yEZ5):** We agreed with reviewers xkVH and yEZ5 that the current paper's title (Discovering Diverse Behaviors via Temporal Contrastive Learning) might be confusing, and we would like to improve the clarity with a better title. We updated the draft’s title to "Temporal Representations for Exploration: Learning Complex Exploratory Behavior without Extrinsic Rewards." Our algorithm learns complex exploratory behavior without relying on extrinsic task information, unsupervised skill learning, or world-modeling. We believe that a new title clarifies the paper's main claim.

We believe the revisions address the main reviewer's concerns. We added a comparison to a model-based exploration baseline in Appendix L. **All changes of the paper are highlighted in blue.**

---

### Meta-Review · Area_Chair_ewNS · 2026-01-01

**Summary:**

The paper introduces an exploration method based on intrinsic rewards derived from temporal contrastive representations. Theoretical results motivate the proposed approach, which is shown to perform well across a wide range of tasks.

There are, however, some clear ambiguities in the paper. In particular, prior work optimizes the sum of intrinsic and extrinsic rewards, whereas this paper considers as a baseline the same methods optimized using only intrinsic rewards. That said, the overall assessment of the paper has been fairly positive, and the discussion phase was very productive. I therefore recommend that the paper be accepted, with the proposed changes highlighted in blue (including the revised title).

**Reviewer Concerns:**

The reviewers were fairly positive in their initial assessment and the authors nicely summarized their interaction in the last message they sent to the AC.

**Reviewer Scores:**

The reviewers would have kept their score or raised it. As all reviewers already are recommending the papers acceptance, minutia about individual reviewers seems pointless.

---

### Decision · Program_Chairs · 2026-01-26

Accept (Poster)